**eLife** | RESEARCH ARTICLE

# Caffeic acid phenethyl ester protects *Clostridioides difficile* infection by toxin inhibition and microbiota modulation

**Yan Guo[1†], Yong Zhang[2†], Guizhen Wang[3], Hongtao Liu[1], Jianfeng Wang[1], Xuming Deng[1], Liuqin He[4*], Jiazhang Qiu[1*]**

[1]State Key Laboratory for Diagnosis and Treatment of Severe Zoonotic Infectious Diseases, Key Laboratory for Zoonosis Research of the Ministry of Education, College of Veterinary Medicine, Jilin University, Center for Pathogen Biology and Infectious Diseases, The First Hospital of Jilin University, Changchun, China; [2]Center for Pathogen Biology and Infectious Diseases, State Key Laboratory for Zoonotic Diseases, The First Hospital of Jilin University, Changchun, China; [3]Measurement Biotechnique Research Center, College of Biological and Food Engineering, Jilin Engineering Normal University, Changchun, China; [4]Hunan Provincial Key Laboratory of Animal Intestinal Function and Regulation, Hunan International Joint Laboratory of Animal Intestinal Ecology and Health, Laboratory of Animal Nutrition and Human Health, College of Life Sciences, Hunan Normal University, Changsha, China

**\*For correspondence:**
heliuqin@hunnu.edu.cn (LH);
qiujz@jlu.edu.cn (JQ)

[†]These authors contributed equally to this work

## eLife Assessment

This **valuable** study by Guo and colleagues reports the inhibitory activity of caffeic acid phenethyl ester (CAPE) against TcdB, a key toxin produced by Clostridioides difficile. C. difficile infections are a major public health concern, and this manuscript provides interesting data on toxin inhibition by CAPE, a potentially promising therapeutic alternative for this disease. The strength of the evidence to support the conclusions is **solid**, with some concerns about the moderate effects on the mouse infection model and direct binding assays of CAPE to the toxin.

**Abstract** *Clostridioides difficile* infection (CDI) is the leading cause of hospital-acquired diarrhea that seriously threatens public health. The disruption of normal gut microbiota by the use of broad-spectrum antimicrobial agents enables *C. difficile* to proliferate in the colon. The emergence and prevalence of hypervirulent *C. difficile* strains result in increased morbidity, mortality, and high recurrence rates of CDI, thus creating a pressing need for novel therapeutics. The multi-domain toxins TcdA and TcdB are the primary determinants of CDI pathogenesis, rendering them ideal drug targets in the anti-virulence paradigm. In this study, we identified caffeic acid and its derivatives from natural compounds library as active inhibitors of TcdB via a cell-based high-throughput phenotypic screening. Further mechanistic investigations revealed that caffeic acid phenethyl ester (CAPE) could directly bind to TcdB, thus suppressing InsP$_6$-induced autoproteolysis and inhibiting glucosyltransferase activity. CAPE treatment remarkably reduces the pathology of CDI in a murine infection model in terms of alleviated diarrhea symptoms, decreased bacterial colonization, and relieved histopathological lesions. Moreover, CAPE treatment of *C. difficile*-challenged mice induces a remarkable increase in the diversity and composition of the gut microbiota and alterations of gut metabolites (e.g., adenosine, D-proline, and melatonin), which might partially contribute to the therapeutic outcomes of CAPE against CDI. Our results reveal the potential of CAPE as a therapeutic for the management of CDI, or CAPE might serve as a lead compound for the development of antivirulence drugs targeting TcdB.

## Introduction

*Clostridioides difficile* is an obligate anaerobic, spore-forming, gram-positive bacterial pathogen that is the most common cause of hospital-acquired and antibiotic-associated gastrointestinal diseases, including mild diarrhea and life-threatening pseudomembranous colitis (*Smits et al., 2016*; *Miller et al., 2011*; *Kelly and LaMont, 2008*). In the United States, *C. difficile* is responsible for almost half a million infections and approximately 29,000 deaths per year, resulting in an estimated annual healthcare-associated cost of up to $4.8 billion (*Dubberke, 2012*; *Lessa et al., 2015*). Importantly, the incidence, severity, mortality, and healthcare costs of *C. difficile* infection (CDI) have dramatically increased in the 21st century, at least partially owing to the frequent use of antimicrobial agents and the emergence of hypervirulent strains (e.g., PCR ribotypes 027 and 078) (*Goorhuis et al., 2008*; *He et al., 2013*; *Clements et al., 2010*). Limited options are available for the treatment of CDI, while antibiotics including metronidazole, fidaxomicin, and vancomycin are the mainstays of clinical practice (*Cornely et al., 2012*; *Bingley and Harding, 1987*). However, antibiotic treatment is ineffective in more than 35% of CDI patients and can lead to infection recurrence (*Feuerstadt et al., 2023*; *Hopkins and Wilson, 2018*). Moreover, the rapid evolution of antibiotic resistance in *C. difficile* has further compromised the therapeutic efficacy of current treatment options (*Spigaglia et al., 2018*). Therefore, the high morbidity and mortality, high recurrence rates, and emergence of multiple antibiotic-resistant isolates have made this pathogen a major threat to public health; these factors highlight the urgent need for novel therapies.

*C. difficile* widely exists in the environment and in the intestinal tract of humans and animals and is mainly transmitted via the fecal–oral route (*Seekatz and Young, 2014*). The pathogenesis of CDI is correlated with the use of antibiotics, which impair the structure and composition of the normal gut microbiota, thus allowing *C. difficile* to proliferate in the colon (*Seekatz and Young, 2014*). Therefore, previous antibiotic therapy is the major risk factor for the development of CDI. In the colon, the dormant spores of *C. difficile* germinate into oxygen-sensitive vegetative cells that produce toxins and cause symptoms of disease (*Paredes-Sabja et al., 2014*). *C. difficile* secretes two large clostridial exotoxins (TcdA and TcdB), which are directly responsible for the clinical pathology of CDI (*Lyras et al., 2009*; *Kuehne et al., 2010*). Epidemiological studies suggest that most disease-causing *C. difficile* strains have genes that encode TcdB (*Carter et al., 2015*; *Drudy et al., 2007*). TcdA and TcdB have identical functional domains: a N-terminal glucosyltransferase catalytic domain (GTD), a cysteine protease active domain (CPD), a transmembrane domain (DRBD), and a C-terminal receptor-binding domain (CROPs) (*Chen et al., 2019*). When secreted into the colon, both toxins are internalized by host cells through receptor-mediated, clathrin-dependent endocytosis. Following a decrease in endosomal pH, the toxins undergo conformational changes in the DRBD, resulting in insertion of the toxin into the endosome membrane, pore formation, and the release of the GTD and CPD domains into the cytosol. Then, CPD binding with cytosolic inositol hexakisphosphate (InsP$_6$) activates its protease activity, leading to the cleavage and release of the GTD. Eventually, the GTD utilizes UDP-glucose to glucosylate Rho and/or Ras family GTPases, thus disrupting GTPase signaling and resulting in deleterious effects, including cytopathic 'rounding', dysregulation of the actin cytoskeleton, apoptosis, and pyroptosis (*Smits et al., 2016*). Although both toxins are cytotoxic to host cells, TcdB shows greater toxicity than TcdA (*Orrell and Melnyk, 2021*).

Disruption of the gut microbiota by antimicrobial agents results in high recurrence rates, which limits the application of antimicrobial therapy for CDI (*Seekatz and Young, 2014*). Consequently, there is a critical need to develop nonantibiotic treatments for CDI, either as standalone or augmented therapies. Importantly, strategies targeting bacterial virulence factors, including toxins, are termed anti-virulence strategies and have attracted interest in the 21st century (*Rasko and Sperandio, 2010*). Owing to its essential role in *C. difficile* pathogenesis, TcdB has become an ideal target for the development of anti-virulence drugs (*Tam et al., 2015*). In this study, caffeic acid and a series of its derivatives were found to inhibit TcdB-mediated cell rounding via cell-based screening. Among these potential inhibitors, caffeic acid phenethyl ester (CAPE) had the greatest inhibitory effect. Mechanistically, CAPE bound to both full-length TcdB and the GTD fragment, leading to alterations in the secondary structure. This interaction suppressed the InsP$_6$-mediated self-cleavage of TcdB, as well as the enzymatic activity of GTD, thereby decreasing the level of glucosylated Rac1 in TcdB-treated host cells and further inhibiting the cytopathology and cytotoxicity of TcdB. Oral administration of CAPE significantly protected against CDI in models of animal infection. In addition to direct toxin inhibition,

the therapeutic effect may also be attributed to CAPE-induced alteration of the gut microbiota (e.g., *Bacteroides* spp.) as well as gut metabolites (e.g., adenosine, D-proline, and melatonin). Overall, our study revealed that caffeic acid and its derivatives, particularly CAPE, may serve as promising lead compounds for the development of anti-virulence drugs to treat CDI.

## Results

### CAPE inhibits cell rounding induced by TcdB

TcdB can inactivate Rho GTPases via glucosylation, leading to the loss of cytoskeletal structure and cell rounding. Consequently, we utilized this phenotype to identify potential TcdB inhibitors from 2076 chemicals, including 1515 FDA-approved drugs and 561 natural compounds. The minimal concentration of recombinant $His_6$-TcdB (0.2 ng/ml) that induced 100% cell rounding within 1 hr was used for drug screening. This concentration of TcdB was preincubated with 8 µg/ml of the individual compounds for 1 hr. The compounds that were not bound to TcdB were removed by ultrafiltration centrifugal partition and then added to Vero cell cultures. Residual compounds that have not bound to TcdB

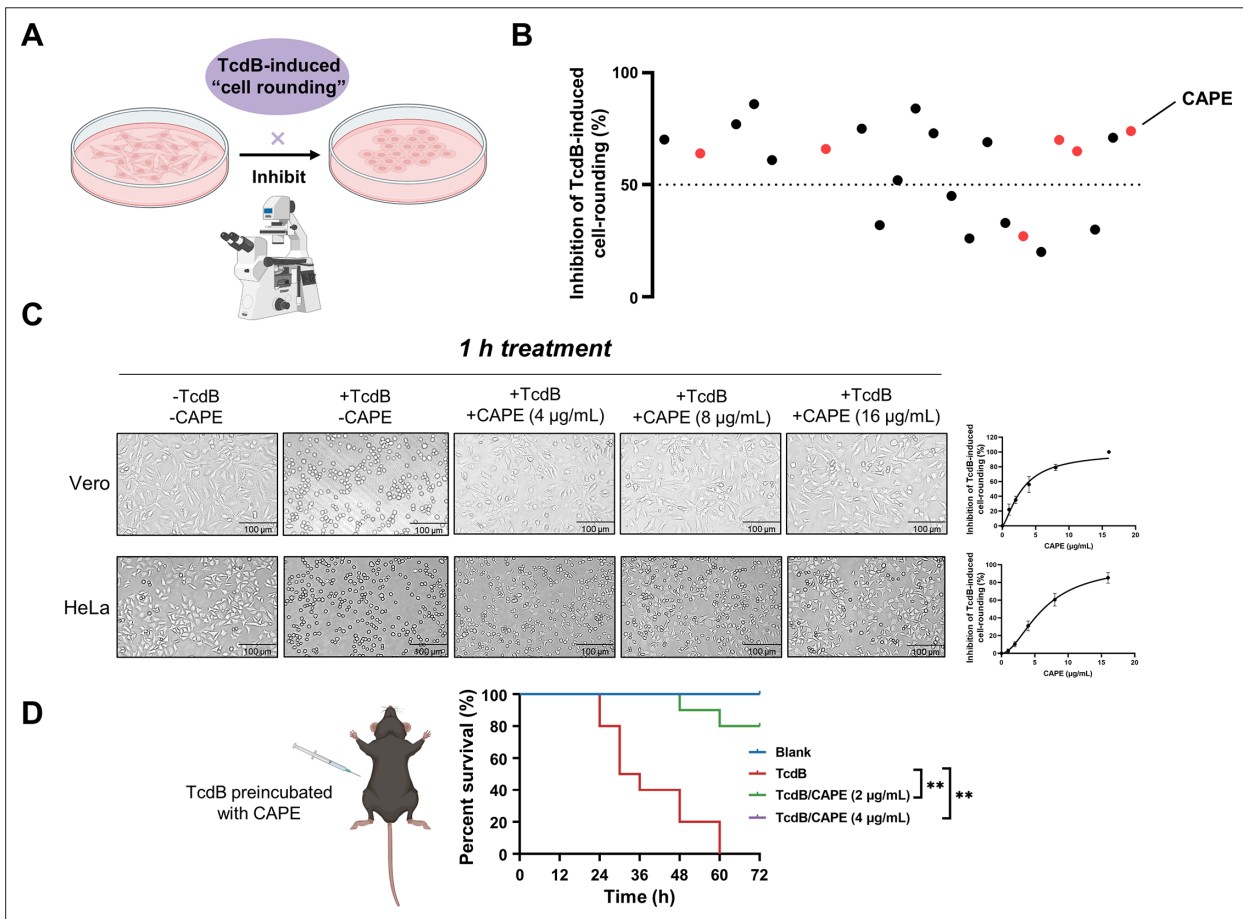

**Figure 1.** Caffeic acid phenethyl ester (CAPE) suppresses TcdB-mediated cell rounding. (**A**) Schematic of the process used for the identification of small molecules that inhibit the cell rounding phenotype induced by TcdB. TcdB (0.2 ng/ml) was preincubated with 8 µg/ml of individual compounds prior to addition to the cell cultures. (**B**) Active compounds were identified from the chemical library consisting of 2076 chemicals. The red dots represent caffeic acid and its derivatives. (**C**) Dose-dependent CAPE inhibition of TcdB-induced rounding of Vero (upper) and HeLa (lower) cells. Scale bar, 100 µm. The dose titration curves are presented on the right (mean ± SD; *n* = 3). (**D**) The survival rates of mice that were intraperitoneally injected with CAPE-pretreated or native TcdB (30 ng) (*n* = 10; **p < 0.01 by log-rank (Mantel−Cox) test).

The online version of this article includes the following figure supplement(s) for figure 1:

**Figure supplement 1.** Caffeic acid phenethyl ester (CAPE) reduces cell death induced by TcdB.

**Figure supplement 2.** Caffeic acid phenethyl ester (CAPE) neutralizes TcdB-mediated acute toxicity in mice.

**Figure supplement 3.** Caffeic acid phenethyl ester (CAPE) inhibits the growth of *C. difficile* BAA-1870 and the expression of TcdB.

may lead to an exaggerated assessment of their efficacy and could also result in misleading positive outcomes. As the effective concentration for cellular interactions may inadvertently be increased, such carryover could distort the determination of the compound's minimum effective concentration. At 1 hr post-TcdB treatment, cell morphology was visualized under a light microscope (*Figure 1A*). These analyses allowed us to identify 22 candidate compounds with potent inhibitory effects (*Figure 1B*). Interestingly, 6 out of the 22 compounds were caffeic acid derivatives. The natural compound library contained eight caffeic acid derivatives, of which methyl caffeic acid and ferulic acid displayed no efficacy. Caffeic acid itself only slightly inhibited TcdB-induced cell rounding, with a 50% inhibitory concentration ($IC_{50}$) of 527.5 µg/ml (1.86 mM) (*Supplementary file 1A*). Notably, except for rosmarinic acid, the $IC_{50}$ values for the other screened caffeic acid derivatives were less than 6 µg/ml, suggesting that these compounds potently inhibited TcdB-mediated toxicity (*Supplementary file 1A*). CAPE is the phenethyl alcohol ester of caffeic acid and a main active component of propolis with broad pharmacological properties, such as anticancer, anti-inflammatory, antiviral, antibacterial, antifungal, antioxidant, and immunomodulatory activities (*Balaha et al., 2021*). As CAPE has the lowest $IC_{50}$ (3.0 µg/ml) among the active caffeic acid derivatives, it was selected as the representative compound for subsequent investigations (*Supplementary file 1A*). After preincubation with increasing concentrations of CAPE, dose-dependent inhibition of cell rounding was observed in both Vero and HeLa cells (*Figure 1C*). The protective influence of CAPE against TcdB-induced Vero cell death remained evident even when the treatment duration was prolonged to 12 hr (*Figure 1—figure supplement 1A, B*).

In addition, we also determined whether inhibition of TcdB by CAPE could lead to loss of toxin function in vivo using a murine systemic intoxication model. A lethal dose of recombinant $His_6$-TcdB (30 ng) was preincubated with either DMSO or CAPE (2 or 4 µg/ml) for 1 hr at 37°C, followed by intraperitoneal injection into mice. Notably, all mice in the DMSO group (*n* = 10) died within 36 hr after toxin injection. However, 100% of mice pretreated with CAPE 4 (µg/ml) toxin survived at the monitoring endpoint (72 hr) (*n* = 10) (*Figure 1D*). Furthermore, we evaluated the efficacy of CAPE following an intraperitoneal injection by administering it 2 hr prior to the injection of TcdB, in order to gauge the survival rate of the mice. The outcomes revealed that a 2-hr pretreatment with CAPE at doses of 30 and 50 mg/kg significantly enhanced the mice's protection rate by 57.1% (*Figure 1—figure supplement 2*). Simultaneously, we assessed the impact of CAPE on *C. difficile* itself. CAPE markedly repressed *tcdB* gene expression at a concentration of 16 µg/ml (*Figure 1—figure supplement 3A*) and inhibited the growth and spore formation of *C. difficile* BAA-1870 at a concentration of 32 µg/ml (*Figure 1—figure supplement 3B, C*). Taken together, these data indicate that CAPE is a potent toxin inhibitor of TcdB and may serve as a promising antimicrobial agent.

## CAPE decreases InsP$_6$-induced autoproteolysis of TcdB and inhibits the enzymatic activity of GTD

TcdB is a multi-domain containing toxin that induces toxicity in host cells via an intricate and multi-step mechanism. This mechanism involves (1) receptor-mediated endocytosis, (2) pore formation and subsequent membrane translocation, (3) autoproteolysis and release of GTD, and (4) the glucosylation of small GTPases (*Figure 2A*). To determine the mechanism by which CAPE suppresses the cytotoxicity of TcdB, we first labeled full-length TcdB with Alexa Fluor 488; then, the concentration of the endocytosed toxin was determined via flow cytometry. Apparently, incubation of CAPE with TcdB did not significantly affect its entry into host cells (*Figure 2B*). Next, we stained CAPE-treated cells with LysoTracker, a fluorescent dye that accumulates in the acidic organelles of live cells. In contrast to the vacuolar-type H+-ATPase (V-ATPase) inhibitor bafilomycin A1 (Baf-A1), CAPE treatment did not alter the intensity of LysoTracker fluorescence staining in either THP-1 or Caco-2 cells, suggesting that inactivation of TcdB by CAPE is not due to changes in the pH of endosomal compartments (*Figure 2C, D* and *Figure 2—figure supplement 1A, B*).

These observations indicate that CAPE might exert its effects after membrane translocation. Indeed, preincubation of full-length TcdB with CAPE blocked the InsP$_6$-induced autoproteolysis of TcdB, as revealed by the gradual decrease in the production of released GTP, followed by the addition of increasing amounts of CAPE in vitro (*Figure 2E, F*). Furthermore, we determined the glucosyltransferase activity using recombinant GTD in the absence or presence of different doses of CAPE. Strikingly, incubation with 2 µg/ml CAPE resulted in a significant reduction in GTD activity, whereas 32 µg/ml CAPE compromised 87.5% of the enzymatic activity (*Figure 2G*). Importantly, in response

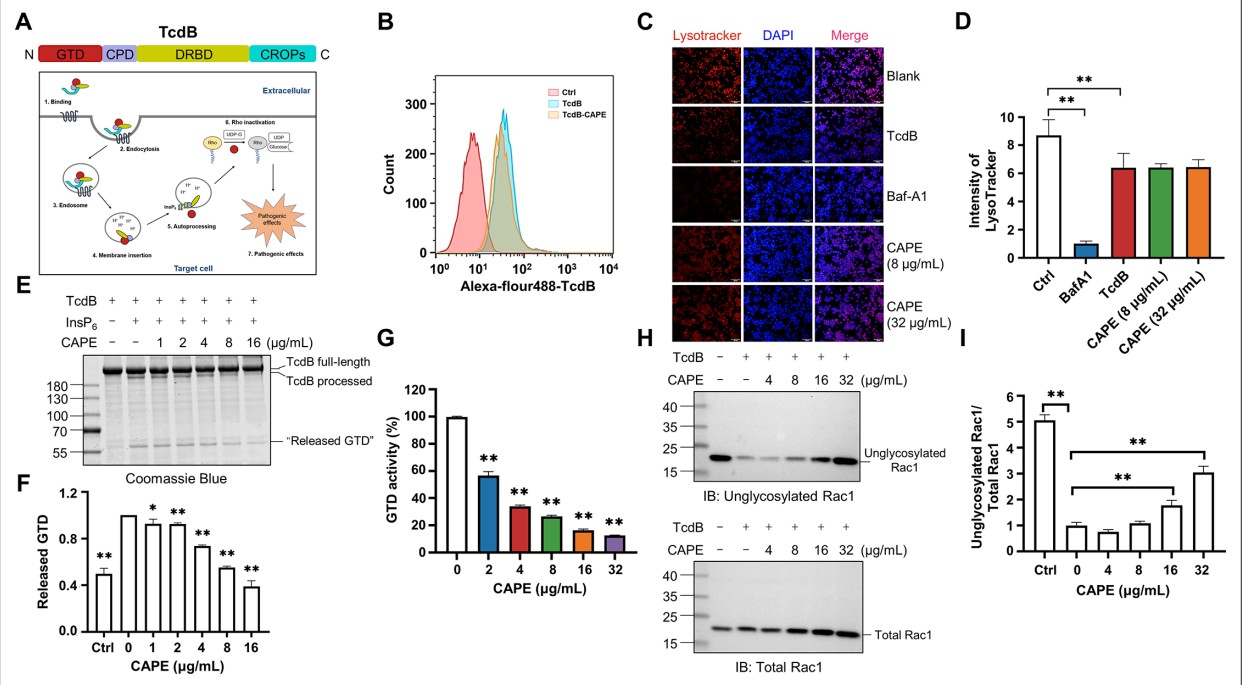

**Figure 2.** Mechanism by which caffeic acid phenethyl ester (CAPE) suppresses TcdB intoxication. (**A**) Schematic illustration of the multistep mechanism of TcdB intoxication of host cells. (**B**) Determination of internalized TcdB levels. Alexa Fluor 488-labeled TcdB was incubated with 32 μg/ml CAPE for 30 min before it was added to the cell culture. The cellular uptake of TcdB was measured by flow cytometry analysis. (**C**) Lysosomal activity in host cells treated with CAPE was measured via LysoTracker staining. The nuclei were stained with DAPI. Bar, 100 μm. (**D**) The intensities of LysoTracker staining were quantified using ImageJ (mean ± SD; $n = 3$; **$p < 0.01$ using one-way ANOVA). (**E, F**) CAPE inhibits InsP$_6$-induced autoprocessing of full-length TcdB. Two micrograms of TcdB was preincubated with increasing concentrations of CAPE for 1 hr. Self-cleavage of TcdB was initiated by the addition of 3 μg of InsP$_6$, and the cells were incubated for 6 hr at 37°C. Reactions were terminated with 5× SDS loading buffer, and the release of GTD was visualized by CBB staining of the SDS−PAGE gel (**E**). The level of released GTD in panel E was quantified by ImageJ (**F**) (mean ± SD; $n = 3$; *$p < 0.05$; **$p < 0.01$ using one-way ANOVA). (**G**) Dose-dependent inhibition of the GTD activity of TcdB by CAPE. Recombinant GTD of TcdB was mixed with different concentrations of CAPE (0–32 μg/ml) and incubated for 15 min at 25°C. GTD activity was measured by the UDP-Glo glycosyltransferase assay (mean ± SD; $n = 3$; *$p < 0.05$; **$p < 0.01$ using one-way ANOVA). (**H, I**) Intracellular Rac1 glucosylation in cells treated with TcdB. Native or CAPE-pretreated TcdB was added to the cell culture and cells were incubated for 2 hr. The cells were lysed, and the glucosylation of Rac1 was analyzed by western blotting using antibodies recognizing total Rac1 and unglycosylated Rac1 (**H**). The level of Rac1 glucosylation shown in H was quantified by ImageJ (**I**) (mean ± SD; $n = 3$; **$p < 0.01$ using one-way ANOVA).

The online version of this article includes the following source data and figure supplement(s) for figure 2:

**Source data 1.** PDF file containing original western blots for **Figure 2** indicating the relevant bands and treatments.zip.

**Source data 2.** Original files for western blot analysis displayed in **Figure 2**.

**Figure supplement 1.** Lysosomal activity of Caco-2 cells in the presence of caffeic acid phenethyl ester (CAPE).

to suppression of InsP$_6$-induced autoproteolysis and GTD function, treatment of TcdB-challenged cells with CAPE markedly decreased the level of glucosylated Rac1 in a dose-dependent manner (**Figure 2H, I**). Taken together, these results suggest that CAPE disrupts InsP$_6$-induced autoproteolysis of TcdB and inhibits the enzymatic activity of GTD, thereby decreasing the glucosylation of Rac1 and alleviating cell rounding.

## CAPE alters the secondary structure of full-length TcdB and GTD

To further clarify the mechanisms used by CAPE to inactivate TcdB, we determined the secondary structures of full-length TcdB and GTD in the absence or presence of CAPE by circular dichroism (CD) spectroscopy. Apparently, incubation of CAPE with full-length TcdB resulted in a significant increase in the α-helix content (15.4–21.0%) and a concomitant decrease in the β-pleated sheet content (34.5–20.9%) (**Figure 3A, B**). Similarly, the secondary structure of GTD was markedly altered upon incubation with CAPE, with the proportion of α-helices decreasing from 37.9% to 0.0% and the proportion of β-pleated sheets decreasing from 37.2% to 10.9% (**Figure 3C, D**). Taken together, these data

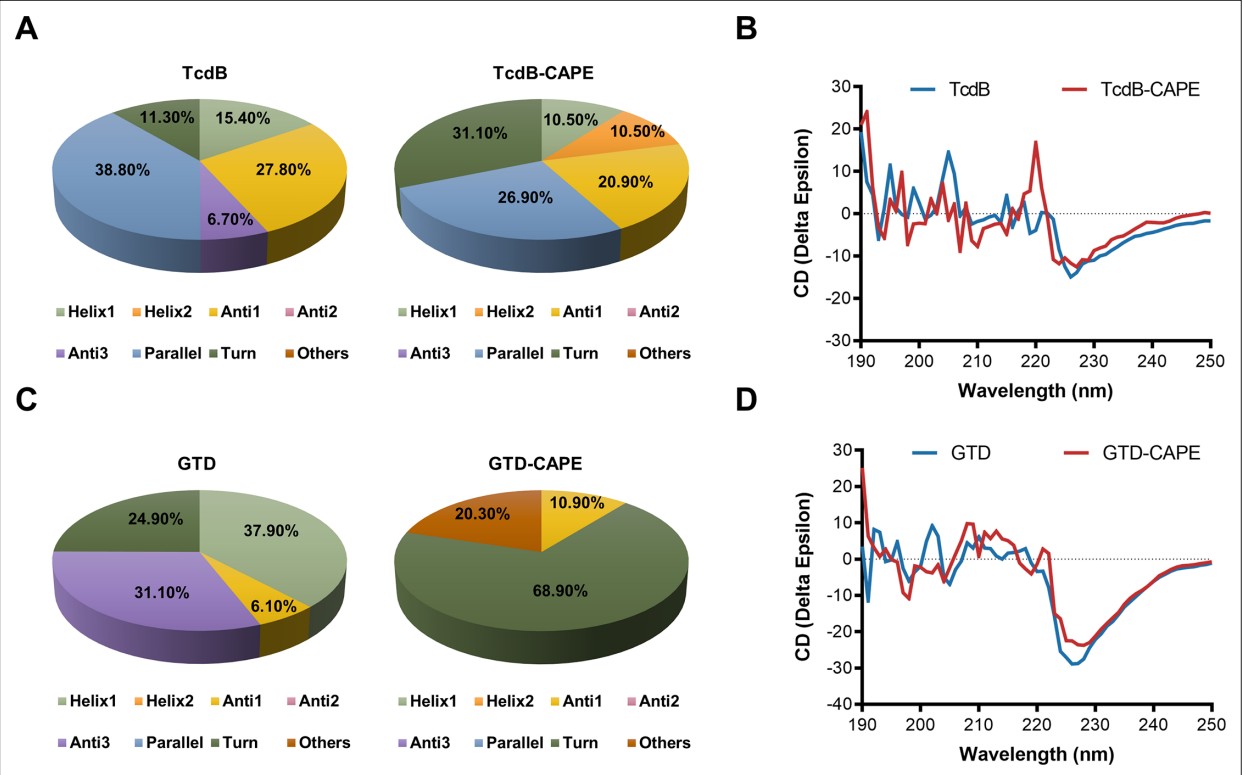

**Figure 3.** Caffeic acid phenethyl ester (CAPE) alters the secondary structure of TcdB and GTD. (**A**) The secondary structure of TcdB in the presence or absence of 32 µg/ml CAPE was determined by circular dichroism (CD) spectroscopy. (**B**) Calculated CD spectra of CAPE-treated TcdB. (**C**) The secondary structure of GTD in the presence or absence of 32 µg/ml CAPE was determined by CD spectroscopy. (**D**) Calculated CD spectra of CAPE-treated GTD. The wavelength for CD spectroscopy was set at 190–250 nm.

suggest that CAPE may interact with full-length TcdB and GTD to induce conformational changes, thus impairing their functions.

## CAPE directly interacts with full-length TcdB and GTD

To investigate the potential interaction between CAPE and TcdB or GTD, the binding constants were determined by surface plasmon resonance analysis. The results showed that CAPE has strong binding affinity with full-length TcdB and GTD, and the KD values were $1.269 \times 10^{-5}$ and $1.007 \times 10^{-3}$ M, respectively (*Figure 4A, B*). The interaction between CAPE and TcdB exhibits a slow approach to equilibrium, which may primarily be attributed to the rapid occupation of the protein's primary binding sites by the small molecule during the initial phase. Following this, CAPE proceeds to bind to sites with secondary or reduced affinity, a process that prolongs the time required to achieve equilibrium. Furthermore, the possibility that CAPE binds to multiple sites on TcdB necessitates a duration for the exploration and occupation of these various sites before equilibrium is established. These data demonstrate the direct interaction between CAPE and full-length TcdB/GTD.

Next, we performed molecular docking and molecular simulation dynamics analyses to investigate the interaction between CAPE and GTD (*Figure 5A*). The results revealed that Leu265, Asp286, Thr465, Ile466, Leu519, and Trp520 in GTD were potential binding sites for CAPE (*Figure 5A*). The root mean square deviation (RMSD) of the GTD/CAPE complex was stable over a 100 ns period of simulation (*Figure 5B*). Further energy decomposition analysis showed that the side chains of Leu265, Thr465, Leu519, and Trp520 of GTD contributed to binding with CAPE via van der Waals interactions, among which Trp520 had the highest binding energy (*Figure 5C*). Moreover, we mutated each of the residues that were involved in binding and subsequently determined their glucosyltransferase activities in the presence or absence of CAPE. The enzymatic activities of these mutants were similar to that of wild-type GTD (*Figure 5D*). Mutations in GTD_L265A, GTD_T465A, GTD_I466A, and GTD_L519A did not affect

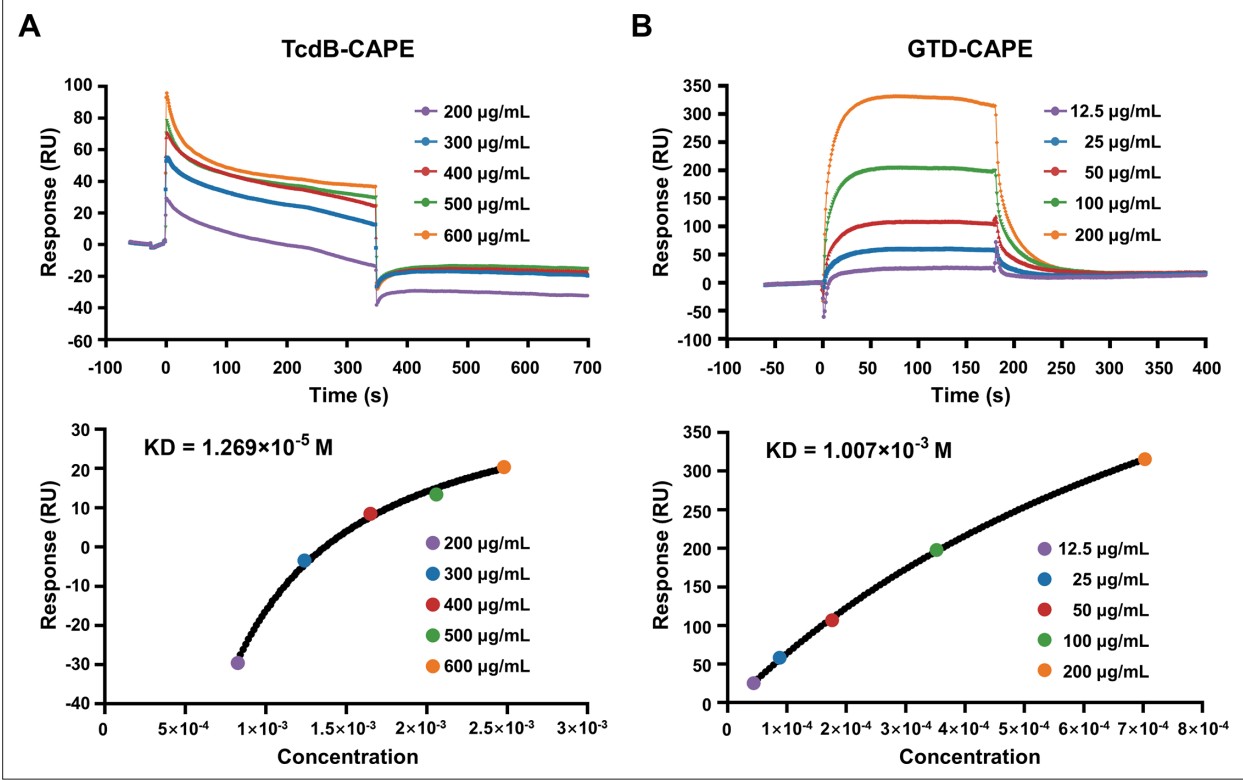

**Figure 4.** Direct interaction of caffeic acid phenethyl ester (CAPE) with TcdB and GTD. (**A, B**) Full-length TcdB or GTD was immobilized on a CM5 chip, followed by the injection of various concentrations of CAPE. The response unit (RU) value and affinities of CAPE interactions with TcdB (**A**) and GTD (**B**) were determined by a Biacore T100.

the inhibitory effect of CAPE (*Figure 5E–I*). However, the inhibition of GTD$_{W520A}$ and GTD$_{D286A}$ by CAPE was notably compromised (*Figure 5J, K*).

## CAPE treatment reduces the pathology of CDI in a murine model

The abovementioned observation that CAPE suppresses the function of TcdB prompted us to further evaluate the potential therapeutic effects of this compound in CDI. To this end, we employed a well-established murine model in which infected mice can develop symptoms resembling those observed in human CDIs (*Garland et al., 2020*). First, cytotoxicity measurements revealed that the viability of both THP-1 and Vero cells was not affected by CAPE treatment (*Figure 6—figure supplement 1A, B*), thus indicating low toxicity. Mice were preconditioned with drinking water containing 0.5 g/l cefoperazone for 5 days, followed by normal drinking water for 2 days. After challenge with *C. difficile* BAA-1870 via gavage, mice were then treated daily with either vehicle or 30 mg/kg CAPE by gavage throughout the disease course (days 0–5) (*Figure 6A*). Severe weight loss in *C. difficile*-infected mice was observed on days 2 and 3 post-infection and was accompanied by diarrhea (*Figure 6B*). Notably, compared to the infection control group, CAPE treatment significantly increased the dry/wet weight ratio of the mouse feces, suggesting the alleviation of diarrhea symptoms (*Figure 6C*). Correspondingly, the weight loss of mice was relieved upon receiving CAPE (*Figure 6B*). Moreover, fecal pellets collected on day 3 were used to determine the bacterial burden. The results revealed that CAPE treatment significantly decreased the number of *C. difficile* colony-forming units (CFUs) (*Figure 6D*). On day 3 post-*C. difficile* infection, the colon was collected for histopathological analysis. Remarkably, *C. difficile*-infected mice exhibited conspicuous colon edema, epithelial cell injury, and inflammatory factor infiltration (*Figure 6E*). In contrast, these histopathological lesions were significantly reduced in mice given CAPE, but pathological changes were still present (*Figure 6F*). This indicates that CAPE may have a limited capacity to reverse the histological damage induced by *C. difficile*. The residual pathology could be attributed to the complex interplay between the pathogen, the host immune

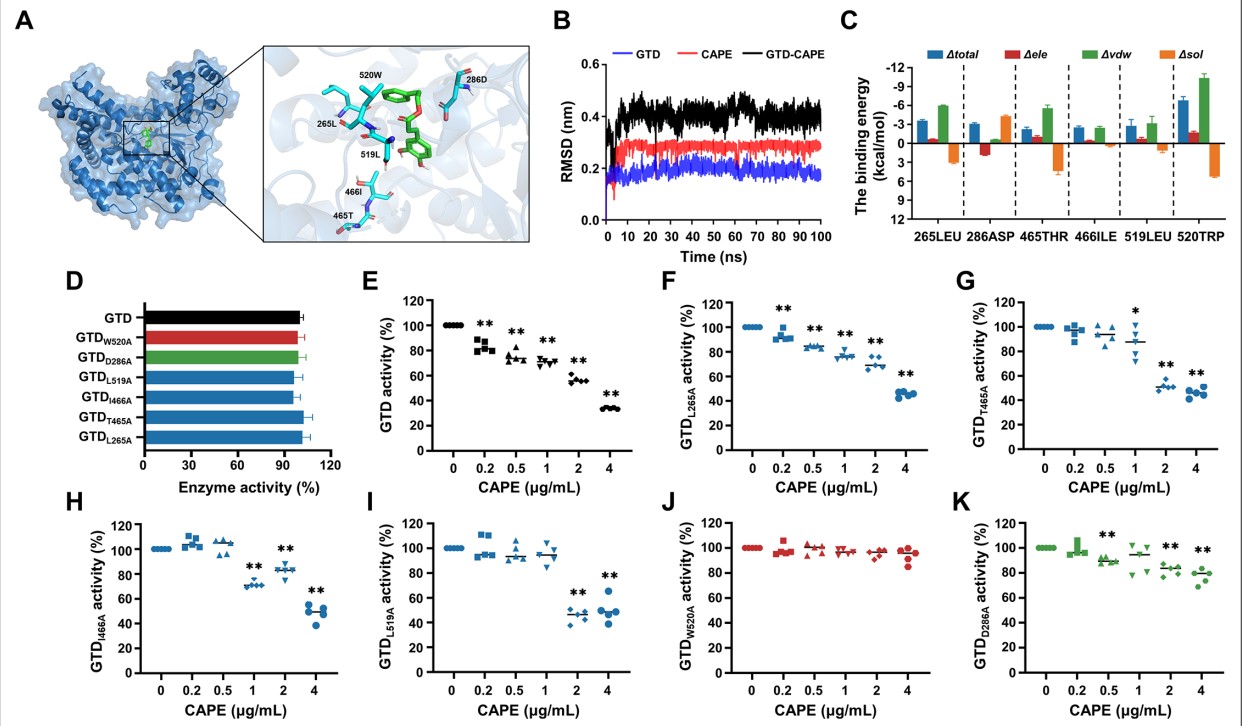

**Figure 5.** Asp286 and Trp520 in the GTD are critical for caffeic acid phenethyl ester (CAPE)-mediated inhibition of GTD. (**A**) Three-dimensional structure determination of GTD with the CAPE complex was performed by molecular modeling. (**B**) The root mean square deviation (RMSD) values of the GTD–CAPE complex. (**C**) Decomposition of the binding energy on a per-residue basis in the binding sites of the GTD–CAPE complex. Blue denotes the total binding energy, red signifies the electrostatic interactions, green corresponds to the van der Waals forces, and orange indicates solvation or hydration effects. The horizontal axis represents the mutation of the amino acid residue at the respective position to alanine (mean ± SD; $n$ = 3). (**D**) Glucosyltransferase activity of wild-type GTD and the GTD mutants (mean ± SD; $n$ = 5). Residual activity of wild-type GTD (**E**) and its mutants GTD$_{L265A}$ (**F**), GTD$_{T465A}$(**G**), GTD$_{I466A}$(**H**), GTD$_{L519A}$(**I**), GTD$_{W520A}$(**J**), and GTD$_{D286A}$(**K**) in the presence of increasing concentrations of CAPE (0–4 µg/ml) (mean ± SD; $n$ = 3; *$p$ < 0.05; **$p$ < 0.01 using one-way ANOVA).

response, and the tissue repair process, which may not be fully mitigated by the inhibitory effects of CAPE on TcdB.

## CAPE induces distinct microbiota changes in CAPE-treated mice when compared with control mice

Antibiotic-induced dysbiosis of the gut microbiota promotes the development of CDI (***Garland et al., 2020***). Therefore, we next determined whether CAPE has any impact on gut microbial communities that may contribute to its therapeutic efficacy. To this end, on the third day after the *C. difficile* challenge, fecal samples from the control group and CAPE treatment group were collected and subjected to microbiota analysis via 16S rRNA gene sequencing. Remarkably, as reflected by the Venn diagram, CAPE treatment resulted in significant changes in the microbial composition compared to those in the control group, and only 16.42% of the operational taxonomic units (OTUs) were shared (***Figure 7A***). In addition, both the α- and β diversity of the gut microbiota in CAPE-treated mice were significantly greater than those in control mice. Specifically, the increase in the Chao1 richness estimator of mice in the CAPE treatment group suggested a greater number of microbial species and increased evenness in the fecal samples (***Figure 7B, C***); the increase in the Shannon–Wiener diversity index and the Simpson diversity index reflected a significant increase in microbiota diversity (***Figure 7D, E***). Principal coordinate analysis revealed distinct confidence ellipses between the bacterial communities of the CAPE-treated group and the control group, thereby revealing substantial differences in their microbiota profiles (***Figure 7F***).

A comparative analysis was performed to examine changes at the phylum level following CAPE treatment. *Bacteroidota*, *Firmicutes*, and *Proteobacteria* were the major constituents in both the CAPE

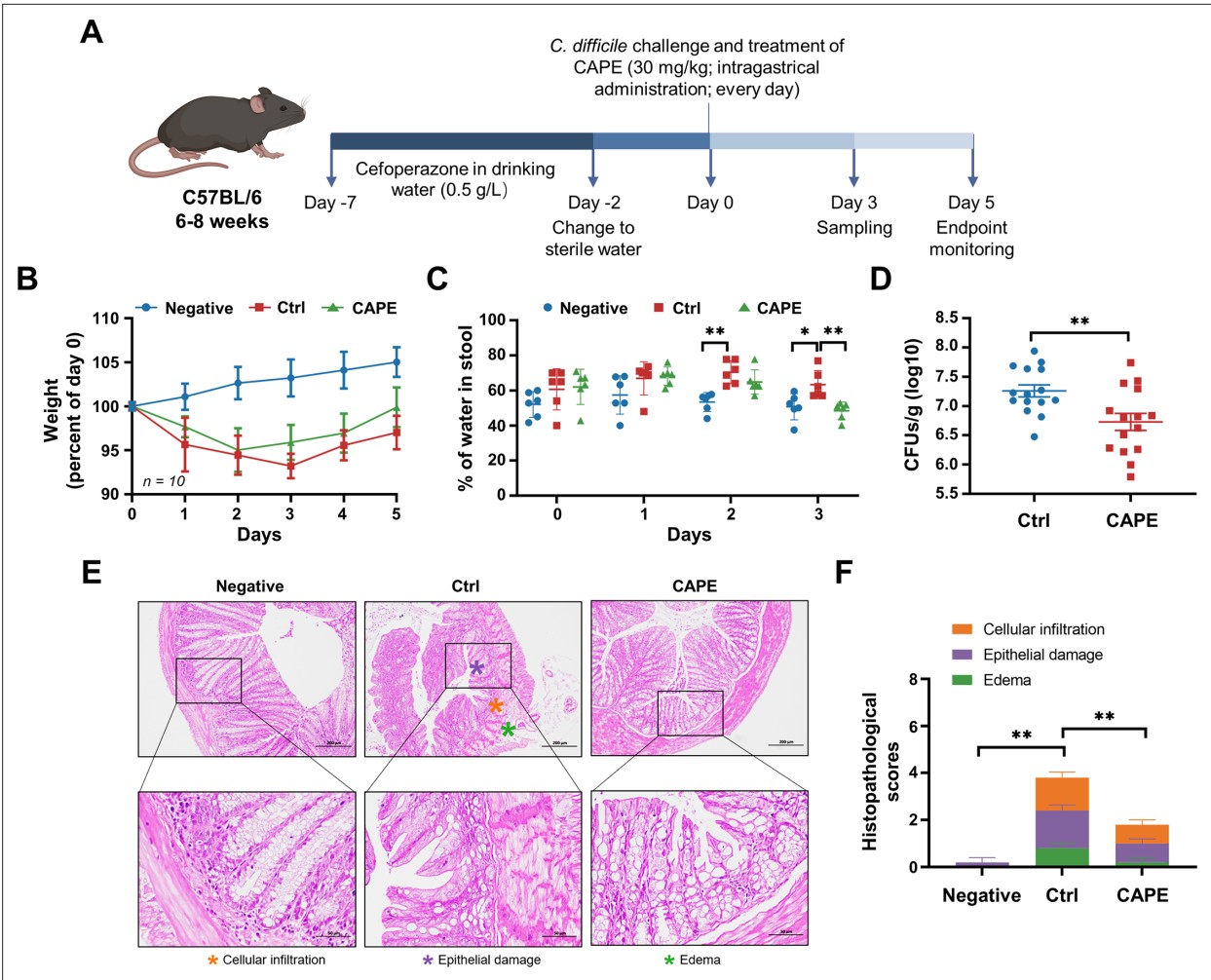

**Figure 6.** Caffeic acid phenethyl ester (CAPE) treatment blocks the pathology of *Clostridioides difficile* infection (CDI) in mice. (**A**) Schematic of the process used for the establishment of the CDI mouse model. (**B**) Weight changes in CDI mice after CAPE or vehicle treatment (mean ± SD; $n = 10$). (**C**) The fecal dry/wet weight ratio in CDI mice treated with CAPE or vehicle (mean ± SD; $n = 6$; *$p < 0.05$; **$p < 0.01$ using one-way ANOVA). (**D**) Bacterial colonization in the feces of CDI mice treated with CAPE or vehicle (mean ± SD; $n = 15$; **$p < 0.01$ using one-way ANOVA). (**E**) Pathological analysis of colon sections from CDI mice treated with CAPE or vehicle. The black boxes are the regions that were magnified and are shown underneath. (**F**) Histopathological scores of vehicle-treated and CAPE-treated mice in terms of cell infiltration (orange), epithelial damage (purple), and edema (green) (mean ± SD; $n = 3$; *$p < 0.05$; **$p < 0.01$ using one-way ANOVA).

The online version of this article includes the following figure supplement(s) for figure 6:

**Figure supplement 1.** Caffeic acid phenethyl ester (CAPE) is not cytotoxic.

treatment and control groups, suggesting that the administration of CAPE did not significantly alter the mouse microbiota at the phylum level (*Figure 7G*). Next, we compared the 10 most abundant taxa at the genus level. Notably, CAPE treatment markedly elevated the abundance of *Bacteroides* and decreased the abundance of *Muribaculaceae* and *Parabacteroides* (*Figure 7H*). Leveraging linear discriminant analysis of effect sizes (LEfSe), the microbiota was examined at the genus level to identify bacteria exhibiting substantial differences in abundance between the two cohorts (LDA score >3.5, p < 0.05). After CAPE treatment, the relative abundance of *Bacteroides*, *Ruminococcus_gnavus_group*, *Muribaculum*, and *Faecalibaculum* OTUs notably increased, while that of *Hungatella*, *Parasutterella*, and *Muribaculaceae* OTUs significantly decreased (*Figure 7I*). *Bacteroides*, *Muribaculum*, and *Faecalibaculum* improve colonization resistance to *C. difficile* (*Li et al., 2021*; *Kienesberger et al., 2021*; *Khanna et al., 2016*). Importantly, the abundance of these bacterial species increased following CAPE treatment (*Figure 7J–L*). In contrast, the abundance of *Parasutterella*, a bacterium that facilitates *C. difficile* colonization (*Dong et al., 2018*), was reduced in mice receiving CAPE (*Figure 7M*). Taken

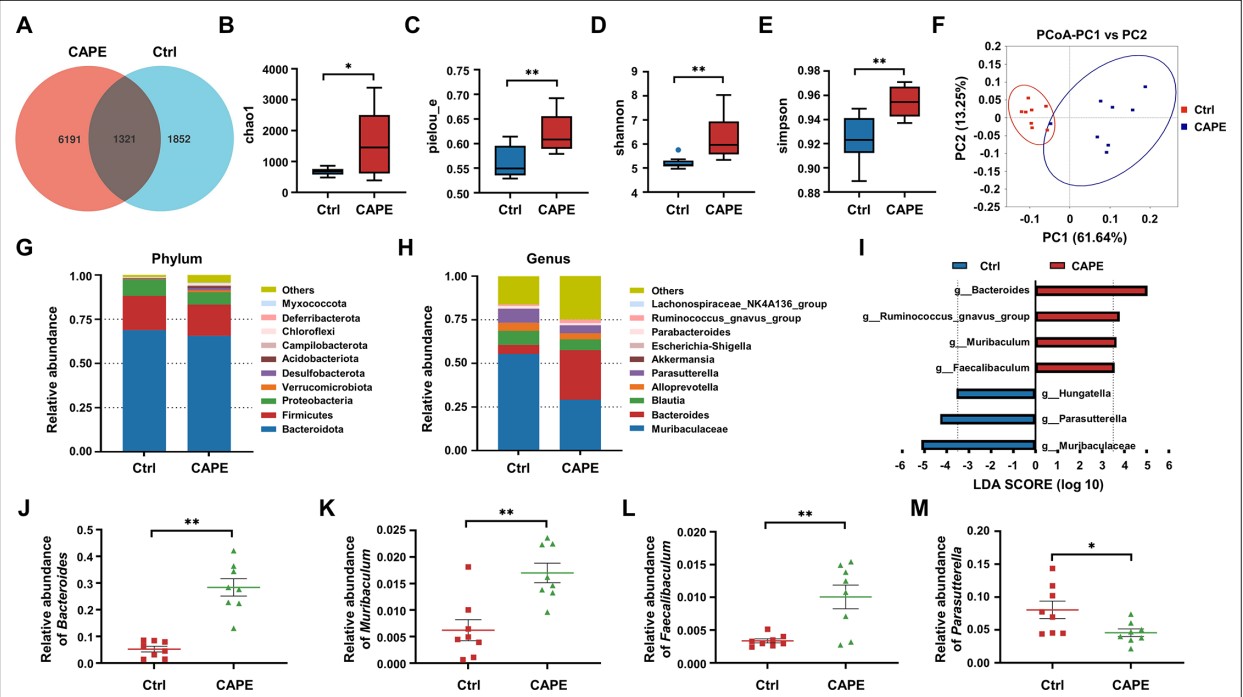

**Figure 7.** The influence of caffeic acid phenethyl ester (CAPE) treatment on the diversity and composition of the gut microbiota in *Clostridioides difficile* infection (CDI) mice. (**A**) Venn diagram illustrating the numbers of differentially abundant bacteria identified using a single model and their cooperation. Quantification of the α diversity of the gut microbiota according to the observed species richness (**B, C**), Shannon index (**D**), and Simpson index (**E**) (mean ± SD; *n* = 8; *p < 0.05; **p < 0.01 using an unpaired Student's *t*-test). (**F**) Principal coordinate analysis (PCoA) showing β diversity based on the weighted UniFrac distance. Relative abundance of the top 10 predominant bacteria classified at the phylum (**G**) and genus (**H**) levels. (**I**) Significant changes in flora among the control and CAPE groups, as measured by LEfSe analysis (LDA score [log 10]>3.5). Relative abundances of *Bacteroides* (**J**), *Muribaculum* (**K**), *Faecalibaculum* (**L**), and *Parasutterella* (**M**) in mice in the control and CAPE treatment groups (mean ± SD; *n* = 8; *p < 0.05; **p < 0.01 using an unpaired Student's *t*-test).

together, these findings suggest that gut microbiota alteration might contribute modestly to the observed effects.

## CAPE alters the levels of gut metabolites in *C. difficile*-infected mice

In addition to the crucial role of the gut microbiota in conferring colonization resistance against *C. difficile*, gut microbiota-derived metabolites significantly contribute to the disease process (*Gurung et al., 2024*). Metabolomic analysis was performed to evaluate the effects of CAPE treatment on gut metabolites in mice. On the third day after infection, the feces of mice in each group were frozen by liquid nitrogen for metabolomics analysis. Specifically, utilizing mass spectrometry-based metabolomics, a simple screening of each metabolite is conducted for parameters such as retention time and mass-to-charge ratio. Subsequently, peak area calibration is performed using the first quality control (QC) sample to enhance the accuracy of identification. The peak extraction is then conducted with settings for mass deviation at 5 ppm, signal intensity deviation at 30%, minimum signal intensity, and adduct ions. Simultaneously, quantification of peak areas is carried out, and target ions are integrated. Molecular formulas are predicted using molecular ion peaks and fragment ions, and these are compared with databases such as mzCloud, mzVault, and Masslist. Background ions are removed using blank samples. The raw quantification results are standardized using the formula: original quantification value of the sample/(total metabolite quantification value of the sample/total metabolite quantification value of QC1 sample), to obtain relative peak areas. Compounds with a coefficient of variation greater than 30% in the relative peak areas of QC samples are excluded. Finally, the identification and relative quantification results of the metabolites are obtained. The partial least square discriminant analysis model exhibited significant separation of clusters between the model and CAPE treatment groups (*Figure 8A*). Among the 1599 metabolites, 109 showed substantial differences in levels between the two groups. Specifically, CAPE-treated mice exhibited 19 metabolites with upregulated

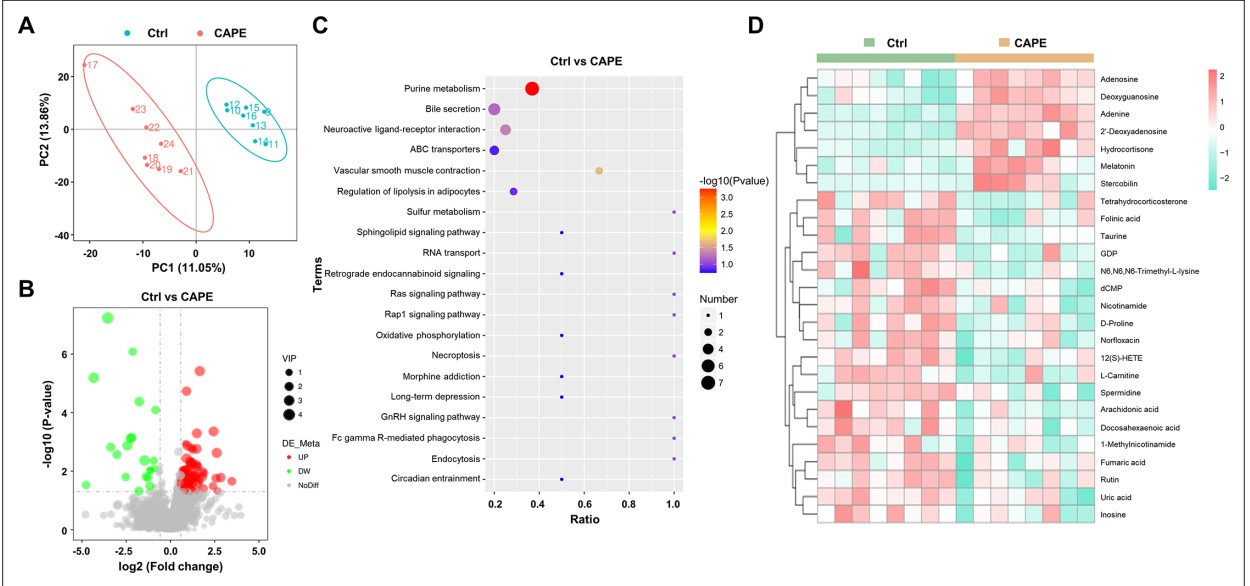

**Figure 8.** Alterations in gut metabolites in caffeic acid phenethyl ester (CAPE)-treated mice with *Clostridioides difficile* infection (CDI). (**A**) The score plots show orthogonal partial least square discriminant analysis (PLS-DA) results. (**B**) Volcano plot representing the up- or downregulated gut metabolites in the CAPE treatment group compared to those in the control group. (**C**) Enrichment bubble map of the top 20 metabolic pathways in the CAPE treatment and control groups. (**D**) The 26 potential biomarkers are displayed in a heatmap, along with hierarchical clustering analysis.

The online version of this article includes the following figure supplement(s) for figure 8:

**Figure supplement 1.** Measurement of melatonin levels in mouse fecal samples.

**Figure supplement 2.** The abundance of melatonin was plotted against the abundance of *Faecalibaculum* based on the microbiota and metabolome profiles.

expression and 50 metabolites with downregulated expression compared to those in the control group (fold change >1.5, p < 0.05) (*Figure 8B*). Furthermore, KEGG enrichment analysis revealed that the differentially abundant metabolites were significantly enriched in the purine metabolism and bile secretion pathways (*Figure 8C*). Adenosine is an endogenous purine nucleoside that binds to specific cell surface receptors to exert anti-inflammatory effects at high concentrations (*Cronstein, 1985*). Importantly, CAPE treatment of *C. difficile*-infected mice can increase the adenosine concentration in the gut (*Figure 8D*), which might be attributed to its inhibition of xanthine oxidase (*Yong et al., 2022*). Similarly, the levels of uric acid, the final product of purine degradation, were markedly decreased in mice after receiving CAPE (*Figure 8D*). An earlier study reported that increasing the gut adenosine concentration with an adenosine deaminase inhibitor prevents *C. difficile* toxin A-induced enteritis in mice (*de Araújo Junqueira et al., 2011*). Therefore, the increase in gut adenosine induced by CAPE might play an important role in its therapeutic efficacy in CDI.

*C. difficile* predominantly utilizes sugars and amino acids for energy production through various metabolic pathways, such as fermentation (*Neumann-Schaal et al., 2019*). Notably, Stickland fermentation, a Clostridia-specific type of amino acid fermentation, is associated with *C. difficile* pathogenesis (*Neumann-Schaal et al., 2019*). D-proline is one of the preferred nutrients used by *C. difficile* in the Stickland reaction and is critical for its growth (*Bouillaut et al., 2013*; *Jackson et al., 2006*). Interestingly, treatment of *C. difficile*-infected mice with CAPE significantly reduced gut D-proline levels (*Figure 8D*). In addition, the metabolomics data also revealed a dramatic increase in the level of melatonin (*Figure 8D*), which is a ubiquitous indolamine that was previously shown to decrease the rate of recurrent CDI (*Sutton et al., 2022*). Consistently, quantification of the melatonin content in mouse fecal supernatants using a melatonin detection kit revealed elevated levels of melatonin in CAPE-treated mice (*Figure 8—figure supplement 1*). Moreover, correlation analysis revealed a positive association between *Faecalibaculum* abundance and melatonin levels (*Figure 1—figure supplement 2*). Taken together, these results indicate that alterations in gut metabolites caused by CAPE treatment might contribute to its therapeutic efficacy in CDI. We found that the metabolites of the gut

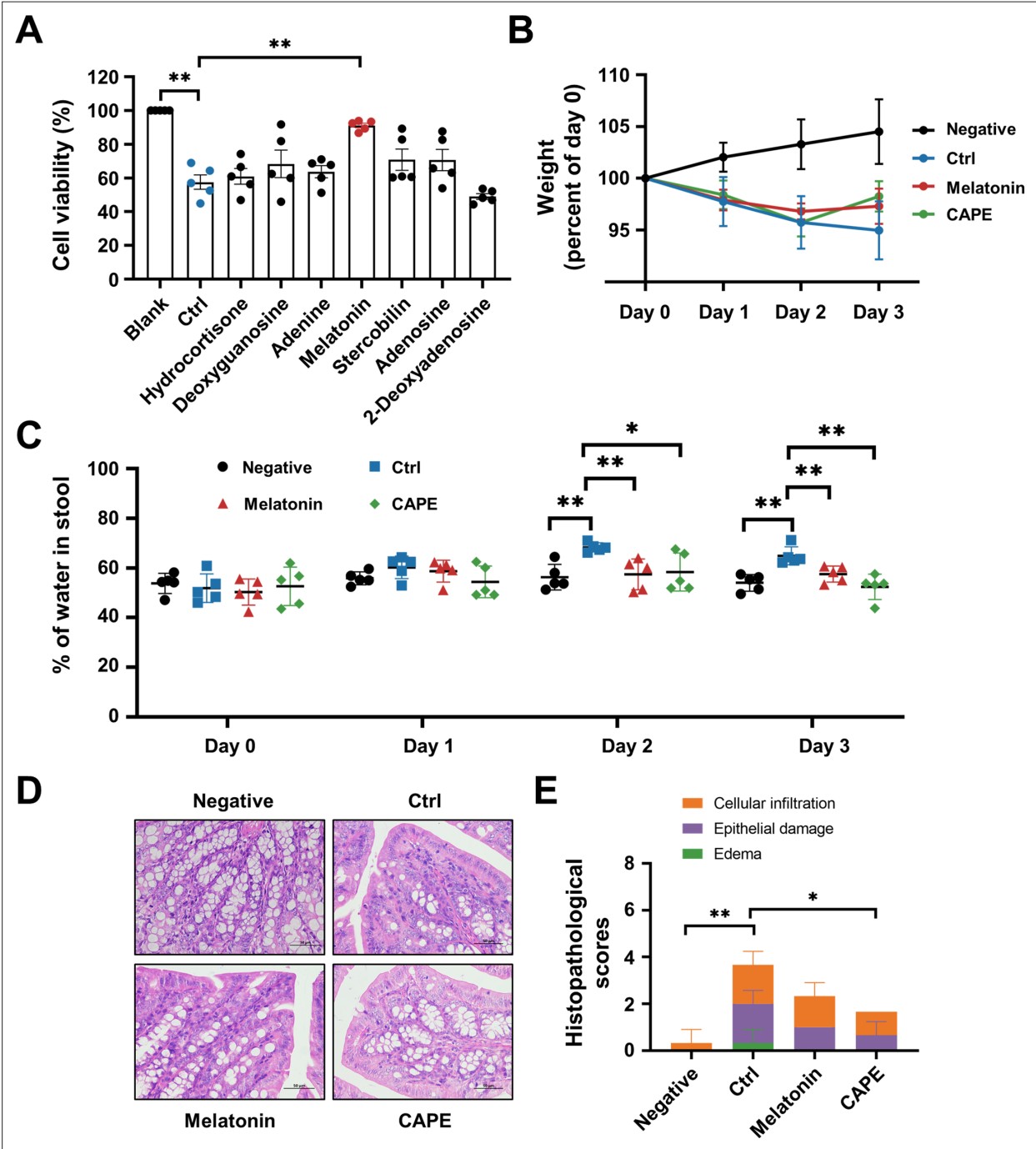

**Figure 9.** Oral administration of melatonin alleviates the pathology of *Clostridioides difficile* infection (CDI). (**A**) The impact of upregulated metabolites on the cytotoxicity of TcdB. TcdB (0.2 ng/ml) was incubated with each of the metabolites (100 μg/ml) for 1 hr at 37°C before being added to the Vero cell culture. At 3 hr after TcdB treatment, cell death was evaluated with a CCK-8 kit (mean ± SD; $n = 3$; **$p < 0.01$ using one-way ANOVA). (**B**) Body weight changes in melatonin-treated CDI mice (mean ± SD; $n = 5$). (**C**) The fecal dry/wet weight ratio in CDI mice after melatonin treatment (mean ± SD; $n = 5$; *$p < 0.05$; **$p < 0.01$ using one-way ANOVA). (**D**) Histopathological analysis of colon tissue from CDI mice treated with melatonin. (**E**) Histopathological scores of melatonin-treated CDI mice (mean ± SD; $n = 3$; *$p < 0.05$; **$p < 0.01$ using one-way ANOVA).

microbiota changed after CAPE treatment. While detectable, the impact of gut microbiota modifications seems to be of secondary importance.

## Oral administration of melatonin alleviates the symptoms of CDI

Next, we investigated whether any of the upregulated metabolites had a direct influence on TcdB. To this end, we preincubated recombinant TcdB with each of the selected metabolites prior to addition to the cell cultures. Interestingly, treatment with TcdB and 100 μg/ml melatonin significantly alleviated the cell death induced by this toxin (*Figure 9A*). This observation prompted us to investigate the direct therapeutic potential of melatonin in CDI. Therefore, *C. difficile*-infected mice were orally administered 200 mg/kg melatonin at 24-hr intervals. The weight changes and the dry/wet ratio of the stool samples were monitored daily for 3 days. The results showed that melatonin-treated mice displayed remarkable alleviation of diarrhea, as manifested by notable improvements in weight recovery and a significant increase in the dry/wet weight ratio on the second and third days after *C. difficile* challenge (*Figure 9B, C*). Consistent with these observations, histopathological assessment of colonic tissue sections also revealed attenuated colon damage upon melatonin treatment (*Figure 9D, E*). Taken together, these data indicate that a high dose of exogenous melatonin could protect mice from CDI. However, the CAPE treatment group induced only minor alterations in gut microbiota-derived melatonin levels, which may not reach therapeutic relevance. While existing studies suggest that melatonin exhibits beneficial effects against CDI, the primary antimicrobial activity appears to be mediated directly by CAPE rather than through melatonin-associated mechanisms.

## Discussion

*C. difficile* has become the prevalent agent of nosocomial diarrhea and substantially threatens public health. In particular, the increase in morbidity and mortality caused by the prevalence of hypervirulent and drug-resistant strains has become a global concern (*He et al., 2013*). In 2019, *C. difficile* was listed by the US Centers for Disease Control and Prevention as an 'urgent threat' that surpassed methicillin-resistant *Staphylococcus aureus* and ESBL-producing *Enterobacterales* (*CDC, 2019*). *C. difficile* is a commensal bacterium that colonizes the lower gastrointestinal tract (*Abbas and Zackular, 2020*). CDI typically occurs in susceptible individuals whose normal gut microbiota is disrupted by the administration of broad-spectrum antibiotics (*Abbas and Zackular, 2020*). Although CDI is considered an antibiotic-associated disease, the current treatment for CDI still relies on antimicrobial drugs, including metronidazole, fidaxomicin, and vancomycin (*Cornely et al., 2012*; *Bingley and Harding, 1987*). Despite antibiotic therapy, more than 35% of CDI patients are susceptible to recurrent infection, which is primarily attributable to the intrinsic effects of antimicrobial agents, which disrupt the microbiota (*Hopkins and Wilson, 2018*). In addition, long-term administration of antibiotics promotes the selection of antibiotic-resistant subpopulations in the gut, thus increasing the risk of infections caused by antibiotic-resistant and opportunistic bacteria (*Raplee et al., 2021*). As a result, the development of nonantibiotic therapeutics is needed for the treatment of CDI.

Indeed, numerous novel treatments for CDI are being investigated, among which the anti-virulence strategy represents one of the most promising and successful therapeutic approaches (*Rasko and Sperandio, 2010*; *Beilhartz et al., 2015*). Unlike antibiotics, which target bacterial components that are critical for bacterial growth, anti-virulence drugs aim to disarm pathogenic bacteria by neutralizing their virulence factors (*Ranftler et al., 2021*; *Zucca et al., 2013*). Therefore, this strategy imposes less selective pressure that directs bacterial evolution of antimicrobial resistance (*Zucca et al., 2013*). The disease progression of CDI is strictly dependent on large clostridial toxins (TcdA and TcdB), as strains lacking both toxins (i.e., A⁻B⁻) were completely avirulent and failed to induce any discernible disease symptoms in a hamster model of infection (*Kuehne et al., 2014*). Importantly, epidemic *C. difficile* strains with increased morbidity and mortality rates produce greater amounts of toxins (*Warny et al., 2005*). Based on these premises, TcdA and TcdB are considered ideal drug targets for CDI in the anti-virulence paradigm (*Stewart et al., 2020*). Indeed, bezlotoxumab, a fully humanized monoclonal antibody that neutralizes TcdB, was approved by the FDA and used in CDI patients to reduce the rate of recurrence (*Deeks, 2017*). This achievement has encouraged researchers to develop next-generation anti-virulence treatments. In particular, small molecule inhibitors of toxin production and/or activity have attracted increasing interest owing to their advantages in cost and the convenience of

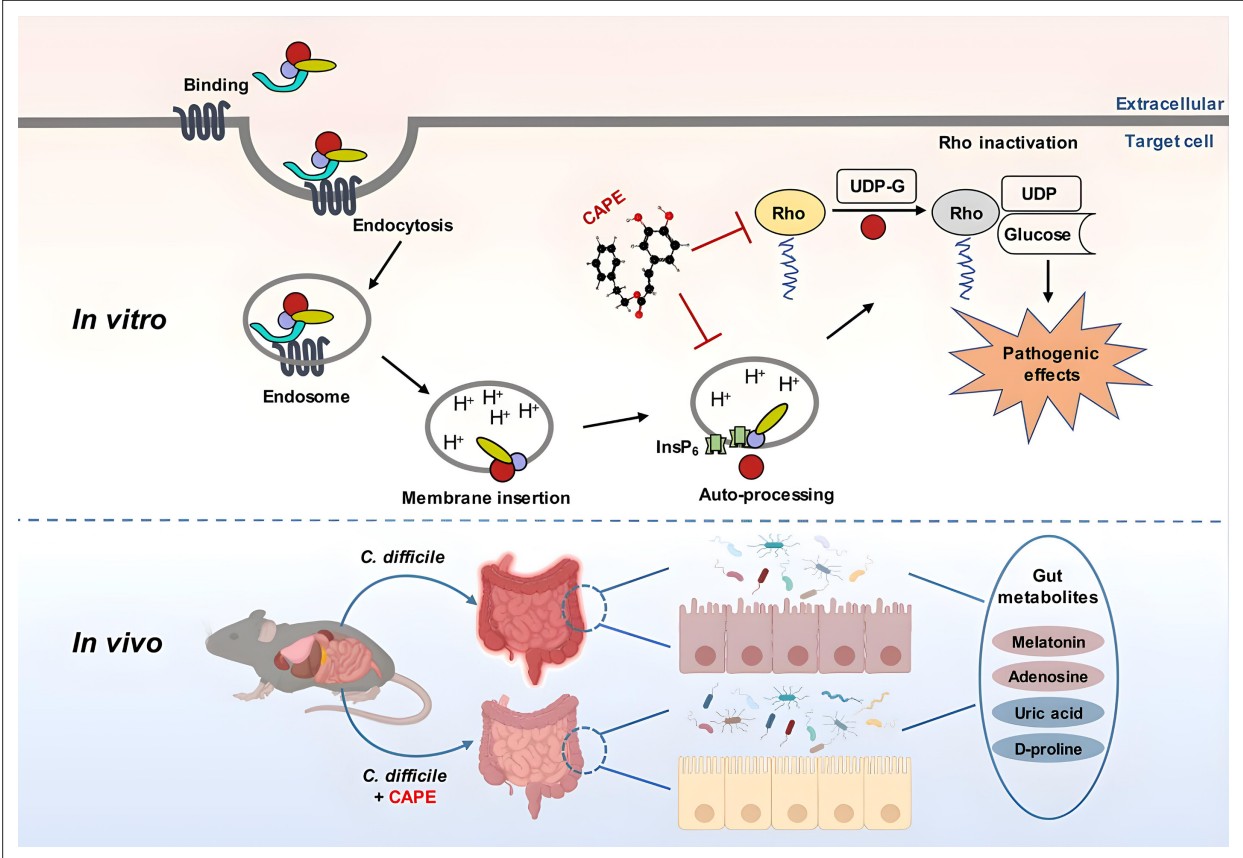

**Figure 10.** A model for caffeic acid phenethyl ester (CAPE)-mediated suppression of TcdB cytotoxicity and protection of mice from *Clostridioides difficile* infection (CDI). The intoxication of cells by TcdB involves multistep mechanisms. Direct interaction of CAPE with full-length TcdB blocks InsP$_6$-induced autoproteolysis, resulting in decreased production of active GTD. The binding between GTD and CAPE inhibits its glucosyltransferase activity, thus decreasing the glucosylation level of Rac1. CAPE treatment of CDI mice markedly restored the diversity and composition of the gut microbiota and induced changes in gut metabolites, which might have contributed to the therapeutic outcomes of CDI following CAPE treatment.

oral administration (*Beilhartz et al., 2015*). The multidomain architecture and mechanisms of action of clostridial toxins have provided distinct options for blocking their toxicity. Indeed, previous efforts have identified diverse inhibitors that act on distinct steps of intoxication, such as the inhibition of toxin binding to cell receptors, the suppression of endosomal maturation, and the inactivation of GTD or CPD activities (*Tam et al., 2015*; *Bender et al., 2015*; *Chan et al., 2022*; *Tam et al., 2018*; *Tam et al., 2020*). Although which approach is the best remains unknown, we believe that an inhibitor that targets multiple steps of intoxication would be superior to those that act on a single step. In this study, we identified a series of caffeic acid derivatives that were potent inhibitors of TcdB via a phenotypic screen. Further mechanistic study revealed that CAPE, the phenethyl alcohol ester of caffeic acid, could block InsP$_6$-induced autoproteolysis of TcdB and inactivate its GTD function (*Figure 10*). CAPE can also inhibit the growth of *C. difficile* and the production of spores. Moreover, CAPE treatment significantly protected against CDI in a mouse model of infection. Our results suggested that caffeic acid derivatives, particularly CAPE, are promising lead compounds for the development of targeted therapies against CDI. CAPE is a major component of propolis, a natural bee-produced substance that has been used as a remedy for thousands of years in Europe and Asia (*Tek et al., 2024*). As a result, further investigations on the therapeutic potential of propolis against CDI, either in combination with standard therapy or as a single therapy, would be of great interest.

Antibiotic-induced gut microbiota dysbiosis is a major risk factor for CDI (*Schubert et al., 2015*). Thus, reconstituting the gut microbiota is a promising strategy for CDI therapy. Fecal microbiota transplantation (FMT) is considered an effective CDI therapeutic strategy and is widely used in clinical practice (*van Nood et al., 2013*). Despite the high cure rate and increasing popularity of FMT, patients remain limited in their acceptance. Another prominent advantage of anti-virulence drugs is

that they allow maintenance of the normal microbiota, which is of critical importance for the treatment of CDI. Importantly, CAPE treatment of *C. difficile*-infected mice restored the structure and composition of the microbiota (*Figure 10*). In particular, the abundance of *Bacteroides*, a critical gut microbiota component that is required for resistance to *C. difficile* colonization (*Li et al., 2021*; *Bornet and Westermann, 2022*; *Pike and Theriot, 2021*; *Deng et al., 2019*), was significantly elevated upon CAPE treatment. Interestingly, *Bacteroides* were also predominant in CDI patients administered RBX2660 (Rebyota), an FDA-approved fecal microbiota product (*Lee et al., 2023*; *Gonzales-Luna et al., 2023*). The increased diversity of the gut microbiota might contribute to the therapeutic effects of CAPE in CDI.

In addition to the microbiota, gut metabolites also play crucial roles in the pathogenesis of CDI, either by promoting or suppressing disease progression (*Gurung et al., 2024*). For example, bile acids and amino acids promote spore germination and pathogenesis, while secondary bile acid and short-chain fatty acids aid in colonization resistance and suppress spore germination (*Gurung et al., 2024*). In addition to the restoration of the gut microbiota, metabolomics analysis of mouse fecal samples revealed dramatic alterations in gut metabolites upon CAPE treatment (*Figure 10*). The differentially abundant metabolites were significantly enriched in the purine metabolism and bile metabolism pathways. Remarkably, the level of adenosine, an endogenous purine nucleoside that can regulate immune responses and inflammatory processes by binding to its receptors, such as its ability to inhibit the release of certain inflammatory mediators, was dramatically increased in CAPE-treated mice (*Haskó et al., 2008*; *Shakya et al., 2019*). In contrast, the amount of uric acid, the final product of purine metabolism, was reduced. Previous studies showed that the increase in adenosine concentrations caused by the inhibition of adenosine deaminase via EHNA prevents *C. difficile* TcdA-mediated damage and inflammation (*de Araújo Junqueira et al., 2011*). This protection might be a result of the anti-inflammatory effects of adenosine, which rely on the activation of its receptors (*Cavalcante et al., 2006*). In addition, we observed a reduction in the amount of D-proline, which is one of the preferred amino acids utilized by *C. difficile* to generate energy through Stickland reactions (*Bouillaut et al., 2013*). Melatonin is a ubiquitous indolamine that has broad biological activities (*Ma et al., 2020*). Notably, we detected a significantly increased amount of melatonin in the feces of mice that were administered CAPE. Importantly, an earlier study demonstrated a decreased risk of recurrent CDI in patients exposed to melatonin, possibly owing to its antibacterial and anti-inflammatory activities (*Sutton et al., 2022*). In this study, we found that oral administration of exogenous melatonin alleviated the symptoms of CDI in a mouse model of infection. Although determining the exact mechanism will require further investigation, we propose that the melatonin-mediated inhibition of TcdB intoxication observed in this study might contribute to its therapeutic effect. Taken together, these results indicate that the alterations in gut metabolites caused by CAPE might also contribute to treatment outcomes for CDI patients.

## Methods

### Cell lines

All cell lines (HeLa, Caco-2, Vero, HEK293T, and THP-1) were obtained from ATCC (Manassas, VA). HeLa (Cat #CCL-2), Caco-2 (Cat #HTB-37), Vero (Cat #CCL-81), and HEK293T (Cat #CRL-3216) cells were cultured in Dulbecco's modified Eagle's medium (DMEM) (Invitrogen, Carlsbad, CA) supplemented with 10% fetal bovine serum (Invitrogen) and 1% (wt/vol) penicillin–streptomycin. THP-1 (Cat #TIB-202) cells were cultured in Roswell Park Memorial Institute (RPMI) 1640 medium (Invitrogen) supplemented with 10% FBS and 1% (wt/vol) penicillin–streptomycin. All cell lines were cultured at 37°C in a 5% $CO_2$ atmosphere. All the purchased cells were identified by STR, and routine mycoplasma contamination testing was conducted every 2 weeks to ensure the reliability of the experimental results.

### Bacterial strains and reagents

*C. difficile* BAA-1870 (RT027/ST1) was kindly provided by Aiwu Wu (Guangzhou Medical University) and cultured in brain–heart infusion (BHI) broth supplemented with 5 g/l yeast extract and 0.1% (wt/vol) L-cysteine (BHIS). The compounds were identified from two main sources: FDA-approved drugs obtained from ApexBio Technology (Houston, TX, USA) and small molecule natural compounds

obtained from Chengdu Ruifensi Biotechnology Co, Ltd (Chengdu, China). LysoTracker Red DND-99, kanamycin, and cefoperazone sodium were purchased from Dalian Meilun Biotechnology Co, Ltd (Dalian, China). Anti-Rac1 antibody Mab102 (1:1000 dilution) was purchased from BD Biosciences (Mississauga, ON), Cat #610651. Anti-total Rac1 antibody 23A8 (used at 1:1000 dilution) was purchased from Millipore (Cat #05-389).

## Mice

Male C57BL/6 mice aged 6–8 weeks were purchased from Liaoning Changsheng Biotechnology Co, Ltd, and housed in a specific pathogen-free (SPF) facility. The mice lived in a controlled environment with a temperature of 23 ± 2°C and a humidity of 55 ± 10%. Prior to infection, the mice were acclimated to the environment for a minimum of 1 week. All mouse experiments were approved by the Jilin University Institutional Animal Care Committee, Jilin University, and strictly conducted according to the guidelines of this committee (SY202402301).

## Protein expression and purification

The expression and purification of full-length TcdB were conducted following the method described by *Yang et al., 2008*. In brief, *Bacillus megaterium* harboring pHis1522-TcdB was inoculated into Luria–Bertani (LB) media supplemented with a final concentration of 10 µg/ml tetracycline. Once the $OD600_{nm}$ reached approximately 0.3, xylose (0.5% wt/vol) was added to induce protein expression at 37°C for 16 hr. Then, 100 ml of the bacterial cells was harvested by centrifugation, resuspended in 5 ml of lysis buffer containing 20 mM imidazole, 300 mM NaCl, 20 mM $NaH_2PO_4$, 500 µM EDTA, and protease inhibitor cocktail (adjusted to pH 8.0), and subjected to sonication. The cell lysate was then centrifuged at 13,400 × *g* for 20 min to remove cell debris. The TcdB protein was purified using nickel column affinity chromatography. The purity of the protein was confirmed by SDS–PAGE, and the protein was subsequently dialyzed against buffer containing 10% glycerol, 20 mM Tris (pH 7.5), and 150 mM NaCl. After the protein concentration was determined, the purified TcdB protein was aliquoted and stored at –80°C for further use.

For the purification of GTD, amplified DNA products were inserted into pET28a (Novagen), and the *Escherichia coli* strain BL21 (DE3) was used for the expression and purification of GTD. Transformed *E. coli* pET28a-GTD was cultured in LB medium supplemented with 10 µg/ml kanamycin. Once the OD600 reached 0.6, the expression of proteins was induced with 0.2 mM isopropyl-β-D-thiogalactoside overnight at 18°C. The bacterial cells were then harvested by centrifugation and subjected to sonication. The cell lysate was then centrifuged at 13,400 × *g* for 20 min to remove cell debris. GTD protein was purified using nickel affinity chromatography and dialyzed in buffer containing 10% glycerol, 20 mM Tris (pH 7.5), and 150 mM NaCl. The GTD mutants were introduced into pET28a-GTD using the Quick Change Site-Directed Mutagenesis Kit (Stratagene, San Diego, CA, USA) with the primers listed in *Supplementary file 1B*. The subsequent purification was performed using the same procedure as described for wild-type GTD.

## Reverse transcription-quantitative PCR

Total RNA was extracted by TRIzol and then reverse-transcribed into complementary DNA by a NovoScript Plus All-in-one 1st Strand cDNA Synthesis SuperMix (E047; Novoprotein, China). mRNA expression of *tcdB* was measured by quantitative PCR with NovoScriptSYBR qPCR SuperMix Plus (E096; Novoprotein, China) using the primers shown in *Supplementary file 1C*. 16S rRNA was used as internal control. The relative mRNA expression levels were calculated by the delta-delta CT method.

## Growth curve assays

Growth curve assay was performed to estimate the influence of CAPE on the growth of *C. difficile* BAA-1870. The overnight culture of BAA-1870 was transferred to fresh BHIS medium and incubated at 37°C in an anaerobic atmosphere. The culture was supplemented with different concentrations of CAPE ranging from 0 to 64 µg/ml and the optical density at 600 nm was monitored at 4-hr intervals to assess the growth of BAA-1870.

## Inhibition of spore outgrowth

The impact of CAPE on the sporulation of *C. difficile* BAA-1870 was assessed using the protocol described by Joshua et al. (*Fletcher et al., 2021*). In brief, BAA-1870 cultures were treated with 8 or 32 µg/ml of CAPE for 48 hr. The treated samples were then harvested, subjected to a double wash with phosphate-buffered saline (PBS), and incubated in a 60°C water bath for 30 min. Subsequently, 20 µl of each spore suspension was plated onto BHIS agar supplemented with 0.1% taurine to facilitate germination. These plates were incubated anaerobically at 37°C for 5 days to allow for spore germination. Following this period, the number of resulting colonies on the plates was enumerated.

## Cell rounding assay

Cell rounding assay was performed according to Kristina et al., with appropriate adjustments (*Bender et al., 2015*). Vero or HeLa cells were seeded in 96-well sterile plates using serum-free DMEM overnight, and the compounds were preincubated with TcdB for 1 hr. After incubation, the TcdB-compound complex was added to cells for 1 hr at 37°C. The cell morphology was visualized under a light microscope.

## LysoTracker assay

The pH of endosomes was determined by LysoTracker Red DND-99 (*Chan et al., 2022*). Caco-2 or THP-1 cells in complete DMEM were seeded in 96-well sterile plates overnight. The media was changed to serum-free DMEM for 1 hr, and CAPE was preincubated with TcdB for 1 hr, followed by addition to the cells and incubation for 2 hr. Baf-A1 served as a control. After incubation, LysoTracker and Hoechst dyes were added, and the cells were incubated for 1 hr. Excess dye was removed by changing the media, and alterations in lysosomal red intensity were documented using a fluorescence microscope.

## In vitro TcdB autoprocessing assay

The self-cleavage ability of TcdB primarily depends on its cysteine protease activity (*Bender et al., 2015*). CAPE (1–16 µg/ml) was preincubated with TcdB for 1 hr, followed by the addition of $InsP_6$ and incubation at 37°C for 6 hr. Subsequently, loading buffer was added for SDS–PAGE analysis, and the results were visualized using Coomassie Brilliant Blue staining.

## Rac1 glucosylation assay

Rac1 glucosylation assay was performed following the method described by John et al. (*Tam et al., 2018*). HEK293T cells were seeded into a 6-well plate. The next day, TcdB and different concentrations of CAPE (4–32 µg/ml) were added to cells. After 2 hr, the cells were harvested, lysed using lysis buffer, and subjected to centrifugation at $13,400 \times g$ for 10 min to collect the supernatant. The collected supernatant was then mixed with loading buffer and subjected to SDS–PAGE followed by western blot analysis. The glycosylation level of Rac1 was evaluated by determining the levels of non-glycosylated Rac1 (BD Biosciences, Mississauga, ON) and total Rac1 (Abcam).

## UDP-Glo UDP-glucose hydrolase assay

GTD activity was assessed according to the manufacturer's instructions (*Tam et al., 2018*). A total volume of 16 µl, consisting of 100 nM GTD, various concentrations of CAPE, and glucosylation buffer, was prepared. Four microliters of UDP-glucose at a final concentration of 25 µM was added to the mixture, which was then allowed to react at room temperature for 15 min. To stop the reaction, 10 µl of the reaction mixture was removed and placed in a white polystyrene 96-well plate containing 10 µl of UDP detection reagent. The reaction took place at room temperature for 1 hr. The luminescence value was measured at a 750-ms integral, and the GTD activity was determined based on these measurements.

## CD spectroscopy

CD spectra of TcdB, TcdB-CAPE, GTD, and GTD-CAPE were obtained using a CD spectrophotometer (MOS-500; Bio-Logic, France). TcdB or GTD at a concentration of 0.2 mg/ml was incubated with 32 µg/ml CAPE. Changes in the secondary structure of TcdB or GTD were assessed at room temperature using a quartz cuvette with an optical path length of 1 mm. The scan wavelength range was set

from 190 to 250 nm with a resolution of 0.2 nm and a bandwidth of 1 nm. The secondary structures of TcdB, TcdB-CAPE, GTD, and GTD-CAPE were analyzed using the BeStSel web server.

## Molecular docking

CAPE was obtained as a ligand from PubChem, and its topology was generated using AmberTools. The crystal structure of GTD was retrieved from the Protein Data Bank (PDB) and processed using AutoDockTools. Docking calculations were performed using Autodock Vina, and molecular dynamics simulations were carried out using Gromacs 2020.6. The RMSD of the receptor and ligand was monitored to assess conformational changes during the binding process. The interaction energy was analyzed using the Poisson–Boltzmann surface area (MM-PBSA) method, which is a molecular mechanics approach.

## Flow cytometry

Flow cytometry was used to quantify the cellular uptake of TcdB via endocytosis. To facilitate detection, full-length TcdB was labeled with Alexa Fluor488 NHS Ester (Yeasen Biotechnology Co, Ltd, Shanghai, China). Prior to cell treatment, Alexa Fluor 488-TcdB was preincubated with CAPE at a concentration of 32 µg/ml for 30 min. The treated solution was then added to Caco-2 cells and incubated for 1 hr at 37°C. Following three washes with PBS, the Caco-2 cells were collected, and the fluorescence signal was measured using flow cytometry.

## Cytotoxicity assay

Vero or THP-1 cells were plated into 96-well plates. The next day, different concentrations of CAPE were prepared by diluting them in DMEM and adding them to the cells. After incubation at 37°C for 3 hr, the viability of cells was assessed using a Cell Counting Kit-8 (CCK-8) (Beyotime Biotechnology Co, Shanghai, China) according to the manufacturer's instructions.

## Biacore analysis

The affinity analysis of TcdB or GTD with CAPE was conducted using a Biacore T100 device (Biacore Inc, Uppsala, Sweden) according to the manufacturer's protocol. A CM5 chip (General Electric Company, GE) was utilized to immobilize the TcdB or GTD proteins. Various concentrations of CAPE were injected, and all steps were performed in running buffer consisting of 5% DMSO in PBS-P20 (GE). Evaluation version 3.1 software provided by Biacore and the 1:1 Langmuir combined model were employed for data analysis.

## Mouse infection model

For the acute toxicity model, after pretreatment of TcdB with CAPE (2 or 4 µg/ml) for 1 hr, the mice were randomly assigned and intraperitoneally injected with a complex of TcdB-solvent or TcdB-CAPE (30 ng of TcdB per mouse). The survival of mice was observed for 72 hr.

For the CDI model, mice were given cefoperazone (0.5 g/l) in their drinking water for 5 days (from days –7 to –2), followed by switching to sterilized water for 2 days. On day 0, the mice were challenged with $1 \times 10^8$ CFUs of *C. difficile* BAA-1870 (Ribotype 027) by oral gavage. For the CAPE or melatonin treatment group, mice were treated with CAPE (30 mg/kg by oral gavage) or melatonin (200 mg/kg by oral gavage) every day after *C. difficile* challenge. The weight loss and fecal state of the mice were monitored daily. Fecal samples were collected on days 0–3 to evaluate the stool water content as a marker of diarrhea. Additionally, fecal samples on day 3 were collected to assess *C. difficile* colonization. Colonic tissues were collected on day 3 post-infection, stained with hematoxylin and eosin, and scored as previously described (*Reeves et al., 2011*).

Fecal samples were collected on day 3 post-infection, frozen in liquid nitrogen, and transferred to the laboratory with dry ice for 16S rRNA sequencing and untargeted metabolomics analysis (Novogene Bioinformatics Technology Co, Ltd, Tianjin, China). The melatonin level in mouse stool supernatant was determined using the Mouse MT (Melatonin) ELISA Kit (Sangon Biotechnology Co, Ltd, Shanghai, China).

## Statistical analysis

Western blot densitometry and LysoTracker fluorescence intensity were measured using ImageJ software (NIH, Bethesda, MD, USA). Statistical analysis was performed using GraphPad Prism 9.0

(GraphPad Software, USA). Unpaired two-tailed Student's *t*-tests were used for comparing two groups, and one-way ANOVA was used for analyzing data from more than three groups. The statistical significance was determined using the log-rank (Mantel–Cox) test (survival rates) in vivo. All the data are expressed as the mean ± SD. * indicates $p < 0.05$ and ** indicates $p < 0.01$.

## Acknowledgements

The authors thank Professor Aiwu Wu (Guangzhou Medical University) for the *C. difficile* BAA-1870 strain and Professor Hanping Feng (Tufts University) for the pHis1522-TcdB plasmid. This work was supported by the National Key Research & Development Program of China (2021YFD1801000), National Natural Science Foundation of China (32200141), the Natural Science Foundation of Jilin Province (20230101142JC and 20240101282JC), and The Fundamental Research Funds for the Central Universities.

## Additional information

### Funding

| Funder | Grant reference number | Author |
| --- | --- | --- |
| National Key Research and Development Program of China | 2021YFD1801000 | Jiazhang Qiu |
| National Natural Science Foundation of China | 32200141 | Yong Zhang |
| Natural Science Foundation of Jilin Province | 20230101142JC | Jiazhang Qiu |
| Natural Science Foundation of Jilin Province | 20240101282JC | Hongtao Liu |
| Fundamental Research Funds for the Central Universities | | Jiazhang Qiu |

The funders had no role in study design, data collection, and interpretation, or the decision to submit the work for publication.

### Author contributions

Yan Guo, Data curation, Formal analysis, Investigation, Methodology, Writing - original draft, Project administration; Yong Zhang, Data curation, Formal analysis, Supervision, Visualization, Methodology, Project administration; Guizhen Wang, Resources, Software, Supervision, Methodology; Hongtao Liu, Conceptualization, Validation, Project administration; Jianfeng Wang, Xuming Deng, Supervision, Validation, Visualization; Liuqin He, Software, Validation, Visualization, Methodology, Writing - review and editing; Jiazhang Qiu, Conceptualization, Supervision, Validation, Visualization, Project administration, Writing - review and editing

### Author ORCIDs

Jianfeng Wang ⬧ https://orcid.org/0000-0001-8311-0894
Jiazhang Qiu ⬧ https://orcid.org/0000-0002-7723-5073

### Ethics

All mouse experiments were approved by the Jilin University Institutional Animal Care Committee, Jilin University and strictly conducted according to the guidelines of this committee (SY202402301).

Reviewer #1 (Public review): https://doi.org/10.7554/eLife.101757.4.sa1
Reviewer #2 (Public review): https://doi.org/10.7554/eLife.101757.4.sa2
Author response https://doi.org/10.7554/eLife.101757.4.sa3

## Additional files

### Supplementary files
MDAR checklist

Supplementary file 1. All the tables involved in the supplementary materials.

### Data availability
All 16S rRNA gene sequencing data generated and analyzed during the current study were deposited in the Sequence Read Archive (SRA) under the BioProject accession number PRJNA1100298. Untargeted metabolomics data are available via Metabolights with the identifier MTBLS9938.

The following datasets were generated:

| Author(s) | Year | Dataset title | Dataset URL | Database and Identifier |
|---|---|---|---|---|
| Guo Y, Zhang Y, Wang G, Liu H, Wang J, Deng X, He L, Qiu J | 2024 | Blocking Toxin Function and Modulating the Gut Microbiota: Caffeic Acid and its Derivatives as Potential Treatments for Clostridioides difficile Infection | https://www.ncbi.nlm.nih.gov/bioproject/?term=PRJNA1100298 | NCBI BioProject, PRJNA1100298 |
| Guo Y, Zhang Y, Wang G, Liu H, Wang J, Deng X, He L, Qiu J | 2024 | Blocking Toxin Function and Modulating the Gut Microbiota: Caffeic Acid and its Derivatives as Potential Treatments for Clostridioides difficile Infection | https://www.ebi.ac.uk/metabolights/editor/MTBLS9938/descriptors | Metabolights, MTBLS9938 |

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
