## [Editor Report · eLife Assessment]

This **valuable** study by Guo and colleagues reports the inhibitory activity of caffeic acid phenethyl ester (CAPE) against TcdB, a key toxin produced by Clostridioides difficile. C. difficile infections are a major public health concern, and this manuscript provides interesting data on toxin inhibition by CAPE, a potentially promising therapeutic alternative for this disease. The strength of the evidence to support the conclusions is **solid**, with some concerns about the moderate effects on the mouse infection model and direct binding assays of CAPE to the toxin.

---

## [Referee Report · Reviewer #1 (Public review)]

Summary:

In this manuscript, Guo and colleagues used a cell rounding assay to screen a library of compounds for inhibition of TcdB, an important toxin produced by Clostridioides difficile. Caffeic acid and derivatives were identified as promising leads, and caffeic acid phenethyl ester (CAPE) was further investigated.

Strengths:

Considering the high morbidity rate associated with C. difficile infections (CDI), this manuscript presents valuable research in the investigation of novel therapeutics to combat this pressing issue. Given the rising antibiotic resistance in CDI, the significance of this work is particularly noteworthy. The authors employed a robust set of methods and confirmatory tests, which strengthen the validity of the findings. The explanations provided are clear, and the scientific rationale behind the results is well-articulated. The manuscript is extremely well written and organized. There is a clear flow in the description of the experiments performed. Also, the authors have investigated the effects of CAPE on TcdB in careful detail, and reported compelling evidence that this is a meaningful and potentially useful metabolite for further studies.

Weaknesses:

Although the authors have made changes to the manuscript to address some of my comments, many of the comments were not satisfactorily addressed. Many of the changes are still superficial, and some concerns still need to be addressed. Important details are still missing from the description of some experiments. Authors should carefully revise the manuscript to ascertain that all details that could affect interpretation of their results are presented clearly.

There is still very little discussion (none, really) in the manuscript about the fact that, because the authors observed a significant effect of CAPE on both bacterial growth and spore production, some of the phenotypes observed can no longer be attributed solely to toxin inhibition.

The details about mass spectrometry are still insufficient. It is still unclear whether metabolite identifications were always based on MS1 or MS2. Instead, several details that are really secondary were included. Authors should be unequivocally clear as to how metabolite identities were obtained. They should also indicate which mass spectrometer was used, and there should be a section in the Materials and Methods describing these experiments.

About the removal of carry-over compounds, the authors stated that ultrafiltration centrifugal partition was used. However, although the authors explained this in detail in their response to reviewers file, the details were omitted from the main text. Authors should clearly state in the manuscript text that "Due to the large molecular weight of TcdB, approximately 270 kDa, we selected a 100 kDa molecular weight cutoff ultrafiltration membrane. The centrifugation was performed at 4000 g for 5 min to eliminate the compounds that did not bind to TcdB."

These are important details which need to be included.

---

## [Referee Report · Reviewer #2 (Public review)]

I appreciate the author's responses to my original review. This is a comprehensive analysis of CAPE on C. difficile activity. It seems like this compound effects all aspects of C. difficile, which could make it effective during infection but also make it difficult to understand the mechanism. Even considering the authors responses, I think it is critical for the authors to work on the conclusions regarding the infection model. There is some protection from disease by CAPE but some parameters are not substantially changed. For instance, weight loss is not significantly different in the C. difficile only group versus the C. difficile + CAPE group. Histology analysis still shows a substantial amount of pathology in the C. difficile + CAPE group. This should be discussed more thoroughly using precise language.

The authors did a good job addressing my concerns regarding the infection model by providing a more accurate descriptions in the Results section for histology. However, the weight loss improvement by CAPE does not look like a significant effect, although it is trending towards improvement. This should be more accurately described.

Another minor concern is that the current Abstract is overstating the amount of disease attenuation. I would replace "remarkably reduces the pathology" with "reduces some of the pathology"

---

## [Author Response]

The following is the authors’ response to the previous reviews

**Public Reviews:**

**Reviewer #1 (Public review):**
Summary:In this manuscript, Guo and colleagues used a cell rounding assay to screen a library of compounds for inhibition of TcdB, an important toxin produced by Clostridioides difficile. Caffeic acid and derivatives were identified as promising leads, and caffeic acid phenethyl ester (CAPE) was further investigated.Strengths:Considering the high morbidity rate associated with C. difficile infections (CDI), this manuscript presents valuable research in the investigation of novel therapeutics to combat this pressing issue. Given the rising antibiotic resistance in CDI, the significance of this work is particularly noteworthy. The authors employed a robust set of methods and confirmatory tests, which strengthen the validity of the findings. The explanations provided are clear, and the scientific rationale behind the results is well-articulated. The manuscript is extremely well written and organized. There is a clear flow in the description of the experiments performed. Also, the authors have investigated the effects of CAPE on TcdB in careful detail, and reported compelling evidence that this is a meaningful and potentially useful metabolite for further studies.Weaknesses:The authors have made some changes in the revised version. However, many of the changes were superficial, and some concerns still need to be addressed. Important details are still missing from the description of some experiments. Authors should carefully revise the manuscript to ascertain that all details that could affect interpretation of their results are presented clearly. For instance, authors still need to include details of how the metabolomics analyses were performed. Just stating that samples were "frozen for metabolomics analyses" is not enough. Was this mass-spec or NMR-based metabolomics. Assuming it was mass-spec, what kind? How was metabolite identity assigned, etc? These are important details, which need to be included. Even in cases where additional information was included, the authors did not discuss how the specific way in which certain experiments were performed could affect interpretation of their results. One example is the potential for compound carryover in their experiments. Another important one is the fact that CAPE affects bacterial growth and sporulation. Therefore, it is critical that authors acknowledge that they cannot discard the possibility that other factors besides compound interactions with the toxin are involved in their phenotypes. As stated previously, authors should also be careful when drawing conclusions from the analysis of microbiota composition data, and changes to the manuscript should be made to reflect this. Ascribing causality to correlational relationships is a recurring issue in the microbiome field. Again, I suggest authors carefully revise the manuscript and tone down some statements about the impact of CAPE treatment on the gut microbiota.

Thanks for your constructive suggestion. We have carefully revised the manuscript according to your suggestions.

**Reviewer #2 (Public review):**
I appreciate the author's responses to my original review. This is a comprehensive analysis of CAPE on C. difficile activity. It seems like this compound affects all aspects of C. difficile, which could make it effective during infection but also make it difficult to understand the mechanism. Even considering the authors responses, I think it is critical for the authors to work on the conclusions regarding the infection model. There is some protection from disease by CAPE but some parameters are not substantially changed. For instance, weight loss is not significantly different in the C. difficile only group versus the C. difficile + CAPE group. Histology analysis still shows a substantial amount of pathology in the C. difficile + CAPE group. This should be discussed more thoroughly using precise language.

Thanks for your constructive suggestion. We have carefully revised the manuscript according to your suggestions.

**Reviewer #3 (Public review):**
Summary:The study is well written, and the results are solid and well demonstrated. It shows a field that can be explored for the treatment of CDIStrengths:Results are really good, and the CAPE shows a good and promising alternative for treating CDI.Weaknesses:Some references are too old or missing.Comments on revisions:I have read your study after comments made by all referees, and I noticed that all questions and suggestions addressed to the authors were answered and well explained. Some of the minor and major issues related to the article were also solved. I am satisfied with all the effort given by the authors to improve their manuscript.

Thanks again for your review.

**Recommendations for the authors:**

**Reviewer #1 (Recommendations for the authors):**
The legend of Figure 3SB is incorrect. It should read "Growth curves of C. difficile BAA-1870 in the presence of varying concentrations of CAPE (0-64 µg/mL)". Also, there is something wrong with the symbols in this figure. I suspect what is happening is that the symbols for the concentrations of 32 and 64 µg/mL are superimposing, but this is a problem because the lower line looks like a closed circle, which is supposed to represent the condition where no CAPE was added. The authors should change the symbols to allow clear distinction between each of the conditions.

Thanks for your constructive suggestion. We have modified the panel and figure legend in Figure 3SB. The concentrations of 32 μg/mL and 64 μg/mL are quite similar, which makes it challenging to differentiate between the corresponding data points on the graph. To enhance clarity, we have utilized distinct colors to help distinguish these closely valued lines as effectively as possible.

Since the authors observed a significant effect of CAPE on both bacterial growth and spore production, their discussion and conclusions need to reflect the fact that the effects observed can no longer be attributed solely to toxin inhibition.

Thanks for your comments. We have modified the corresponding description according to your suggestions.

In lines 43-45, authors state that "CAPE treatment of C. difficile-challenged mice induces a remarkable increase in the diversity and composition of the gut microbiota (e.g., Bacteroides spp.)". It is still unclear to this reviewer why mention Bacteroides between parentheses. Does this mean that there was an increase in the abundance of Bacteroides? If that is the case this needs to be stated more clearly.

Thanks for your comments. Treatment with CAPE indeed significantly increased the abundance of *Bacteroides spp.* in the gut microbiota (Figure 7H-J). However, to avoid ambiguity in the abstract, we have chosen to delete the specific mention of *Bacteroides spp.* within the parentheses.

The modifications made to lines 132-135 still do not address my concern. Authors stated in the manuscript that "compounds that were not bound to TcdB were removed". But how was this done? This needs to be clearly explained in the manuscript. In the response to reviewers document, authors state that this was done through centrifugation. But given that the goal here is to separate excess of small molecule from a protein target, just stating that centrifugation was used is not enough. Did the authors use ultracentrifugation? What were the conditions employed. This is critical so that the reader can assess the degree of compound carryover that may have occurred. Also, authors need to clearly acknowledge the caveats of their experimental design by stating that they cannot rule out the contribution of compound carryover to their results.

Thanks for your comments. We employed ultrafiltration centrifugal partition to remove the unbound small molecule compounds. Due to the large molecular weight of TcdB, approximately 270 kDa, we selected a 100 kDa molecular weight cutoff ultrafiltration membrane. The centrifugation was performed at 4000 g for 5 min to eliminate the compounds that did not bind to TcdB. We have incorporated the relevant methods and discussed the potential impacts on the respective sections of the manuscript.

In line 142, authors added the molar concentration of caffeic acid, as requested. Although this helps, it is even more important that molar concentrations are added every time a compound concentration is mentioned. For instance, just 2 lines down there is another mention of a compound concentration. It would be informative if authors also added molar concentrations here and throughout the manuscript.

Thanks for your comments. In our initial test design, we have utilized the concentration unit of μg/mL. However, during the conversion to μM using the dilution method, some values do not result in neat, whole numbers. For instance, the conversion of 32 μg/mL of caffeic acid phenyl ethyl ester yields 112.55 μM, which appears somewhat irregular when expressed in this manner.

Line 277. For the sake of clarity, I would strongly suggest that authors use the term "control mice" instead of "model mice".

Thanks for your comments. We have modified “model mice” to “control mice” throughout the manuscript.

In line 302, the word taxa should not be capitalized. I capitalized it in my original comments simply to draw attention to it.

Thanks for your comments. We have modified this word.

In the section starting in line 318, authors still need to include details of how the metabolomics analyses were performed. Just stating that samples were "frozen for metabolomics analyses" is not enough. Was this mass-spec or NMR-based metabolomics. Assuming it was mass-spec, what kind? How was metabolite identity assigned? Etc, etc. These are important details, which need to be included.

Thanks for your comments. We have added some metabolomics methods in the corresponding section.

In line 338, the authors misunderstood my original comment. This sentence should read "...the final product of purine degradation, were markedly decreased in mice after...".

Thanks for your comments. We have modified this sentence.

Panels of figure 3 are still incorrectly labeled. The secondary structure predictions are shown in A and C, not A and B as is currently stated in the legend.

Thanks for your comments. We have modified the figure legend in Figure 3.

About Figure 5C, I think the authors for the clarification, but this explanation should be included in the figure legend.

Thanks for your comments. We have added the relevant information to the figure legend.